# Live-cell single particle tracking of PRC1 reveals a highly dynamic system with low target site occupancy

Miles K. Huseyin ⬤ [1] & Robert J. Klose ⬤ [1✉]

Polycomb repressive complex 1 (PRC1) is an essential chromatin-based repressor of gene transcription. How PRC1 engages with chromatin to identify its target genes and achieve gene repression remains poorly defined, representing a major hurdle to our understanding of Polycomb system function. Here, we use genome engineering and single particle tracking to dissect how PRC1 binds to chromatin in live mouse embryonic stem cells. We observe that PRC1 is highly dynamic, with only a small fraction stably interacting with chromatin. By integrating subunit-specific dynamics, chromatin binding, and abundance measurements, we discover that PRC1 exhibits low occupancy at target sites. Furthermore, we employ perturbation approaches to uncover how specific components of PRC1 define its kinetics and chromatin binding. Together, these discoveries provide a quantitative understanding of chromatin binding by PRC1 in live cells, suggesting that chromatin modification, as opposed to PRC1 complex occupancy, is central to gene repression.

[1] Department of Biochemistry, University of Oxford, Oxford, United Kingdom. ✉email: rob.klose@bioch.ox.ac.uk

Eukaryotic DNA is wrapped around histones to form nucleosomes and chromatin that organise and package the genome within the confines of the nucleus. In addition to this packaging role, chromatin and its post-translational modification can also profoundly influence gene transcription[1–3]. Therefore, significant effort has been placed on studying how chromatin-modifying enzymes regulate gene expression. However, in many cases, the mechanisms that enable these enzymes to bind chromatin and identify their appropriate target sites remains poorly understood.

Chromatin-based regulation of gene transcription is typified by the Polycomb repressive system, which is essential for normal gene regulation during animal development[4–9]. This system is comprised of two central histone-modifying protein complexes, Polycomb repressive complex 1 (PRC1) and PRC2. PRC1 is an E3 ubiquitin ligase that mono-ubiquitylates histone H2A on lysine 119 (H2AK119ub1)[10,11]. PRC2 is a methyltransferase that methylates histone H3 at lysine 27 (H3K27me1/2/3)[12–15]. Polycomb complexes bind to their target genes and create Polycomb chromatin domains that are characterised by enrichment of H2AK119ub1, H3K27me3, and the Polycomb repressive complexes themselves[16–19]. Once formed, Polycomb chromatin domains are thought to create structural effects on chromatin that repress transcription by limiting access of transcriptional regulators to gene promoters[20–28]. However, our understanding of the mechanisms that enable Polycomb complexes to recognise and bind to target sites in vivo remains rudimentary and the extent to which structural effects underpin gene repression remains unclear.

Dissecting Polycomb complex targeting mechanisms and their function on chromatin has been challenging because PRC1 and 2 are composed of multiple compositionally diverse complexes. This is exemplified by PRC1, which exists as at least six distinct complexes in mammals. The composition of PRC1 complexes is defined by the PCGF protein (PCGF1-6) that dimerises with either RING1A or RING1B to form the catalytic core. PCGF proteins then interact with auxiliary proteins that regulate catalysis and have unique chromatin binding activities[29–40]. Based on subunit composition, PRC1 complexes are often further separated into canonical and variant forms. Canonical PRC1 complexes form around PCGF2 or 4, and interact with CBX and PHC proteins. The CBX proteins have a chromobox domain that binds to H3K27me3 and therefore canonical PRC1 complexes are readers of PRC2 catalytic activity and occupy target sites enriched for H3K27me3[41–43]. Canonical PRC1 complexes also enable the formation of long-range interactions between Polycomb chromatin domains and create more localised chromatin structures that are proposed to inhibit gene expression[24,25,44–47]. In contrast, variant PRC1 complexes form primarily around PCGF1/3/5/6 and interact with RYBP or YAF2 in place of CBX proteins[31,36,39]. Therefore, variant PRC1 complexes do not recognise H3K27me3 and instead have distinct chromatin binding activities. For example, PCGF1- and PCGF6-PRC1 incorporate auxiliary proteins with DNA binding activity that target these complexes to gene promoters[29,30,33,40,48]. Furthermore, variant PRC1 complexes are highly active E3 ubiquitin ligases and define most H2AK119ub1 in vivo[31,38,49,50].

A series of mechanisms, identified mostly from in vitro biochemical experiments, have been proposed to explain how individual PRC1 complexes bind to chromatin[4,6,51]. The contributions of these mechanisms to chromatin binding in vivo have primarily been examined by ensemble fixation-based approaches like chromatin immunoprecipitation coupled with massively parallel sequencing (ChIP-seq). This has provided a static snapshot of PRC1 complex distribution throughout the genome and perturbation experiments have provided some

information about the mechanisms that enable specific PRC1 complexes to bind chromatin. However, ChIP-seq is blind to kinetics and cannot directly compare and quantitate the chromatin binding activities of individual PRC1 complexes. As such, we currently lack a quantitative model to describe chromatin binding and the function of PRC1 in live cells. This represents a major conceptual gap in our understanding of the Polycomb system and its role in gene regulation.

To begin addressing this, live cell fluorescence recovery after photo bleaching (FRAP) approaches have been used to study chromatin binding for a number of Polycomb proteins in Drosophila[52–54]. These important studies revealed that Polycomb proteins interact with chromatin more dynamically than was expected from in vitro chromatin binding experiments, and similar conclusions have emerged from FRAP experiments in mammalian cell culture systems[55–58]. However, FRAP infers single molecule kinetics from ensemble measurements and therefore can overlook essential chromatin binding behaviours that are only evident when individual molecules are directly observed[59–61]. To overcome this limitation, recent studies have leveraged single-molecule live-cell imaging approaches to shed new light on how previously proposed chromatin binding mechanisms shape PRC2 dynamics[58,62]. Similar approaches have also provided initial descriptions of chromatin binding by CBX proteins, which are specific to canonical PRC1 complexes[43,62]. However, because PRC1 is composed of a diverse set of canonical and variant PRC1 complexes, an understanding of chromatin binding by PRC1 in live cells remains absent, presenting a major barrier to understanding how PRC1 regulates gene expression in vivo.

To overcome this, here we combine genome editing and single particle tracking (SPT) to quantify and dissect chromatin binding of PRC1 in live mouse embryonic stem cells (ESCs). This reveals that PRC1 is highly dynamic, with a small number of molecules displaying stable binding to chromatin. By quantifying absolute PRC1 complex numbers and integrating genomic information, we estimate maximum target site occupancy and discover that most PRC1 target genes are sparsely bound by PRC1 complexes. In dissecting the mechanisms that underpin chromatin binding by PRC1, we discover that interaction between its catalytic core and the nucleosome contributes little to chromatin binding, indicating that the observed binding behaviours of PRC1 must be defined by auxiliary subunits that are specific to individual PRC1 complexes. By systematically characterising chromatin binding and occupancy by individual PRC1 complexes, we reveal how distinct chromatin binding modalities are related to the activity and function of PRC1. Furthermore, using genetic perturbation approaches, we dissect the contribution of canonical and variant PRC1-specific targeting mechanisms to their chromatin-binding activities. Together, these discoveries provide a quantitative understanding of chromatin binding by PRC1 complexes in vivo and have important implications for our understanding of PRC1-mediated gene repression.

## Results

**Live-cell imaging reveals that a small fraction of PRC1 is bound to chromatin.** To study PRC1 in live cells we used CRISPR/Cas9-mediated genome editing to engineer a HaloTag[63] into both alleles of the endogenous *Ring1b* gene in mouse embryonic stem cells (ESCs). We chose RING1B as a proxy for PRC1 complex behaviour because its paralogue RING1A is very lowly expressed in ESCs, it dimerises efficiently with PCGF proteins to form the catalytic core of both canonical and variant PRC1 complexes (Fig. 1a), and its chromatin binding has been studied extensively by fixation-based approaches[30–32,35,39,64,65].

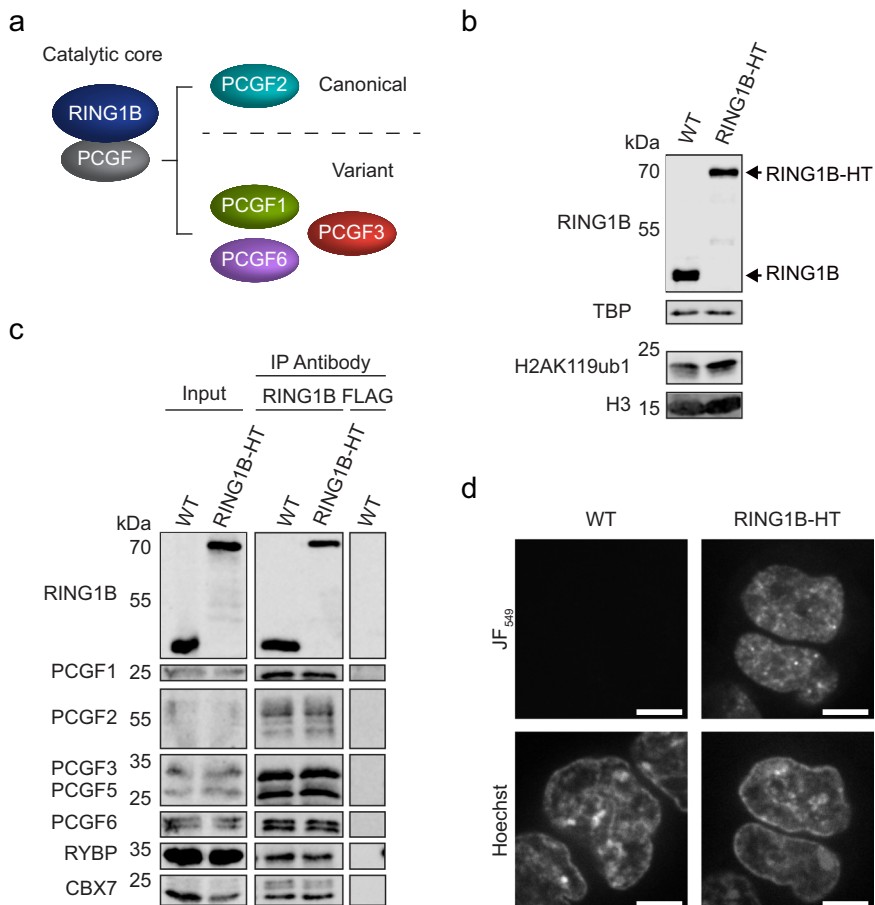

**Fig. 1 Endogenous Halo-tagging of RING1B enables live-cell imaging of PRC1. a** A schematic illustrating the general organisation of PRC1 complexes. **b** Western blots for RING1B (upper panels) and H2AK119ub1 (lower panels) comparing wild type (WT) and homozygous RING1B-HaloTag (RING1B-HT) cell lines. The expected size shift on addition of the HaloTag and linker is 35 kDa. TBP and H3 were used as loading controls. $n = 1$ biological replicate. Source data are provided as a Source Data file. **c** Immunoprecipitation (IP) of RING1B from WT or RING1B-HT ESCs followed by western blotting for PRC1 subunits to confirm normal complex formation. For WT ESCs, a control IP was performed using a FLAG antibody. $n = 1$ biological replicate. Source data are provided as a Source Data file. **d** Representative single Z-slice of untagged (WT) and RING1B-HT ESCs labelled with $JF_{549}$ and Hoechst. Scale bar = 5 μm. Imaging was performed in $n = 2$ biological replicates.

Importantly, addition of a HaloTag to RING1B (RING1B-Halo-Tag) did not alter RING1B expression or H2AK119 ubiquityla-tion (Fig. 1b and Supplementary Fig. 1a). Furthermore, biochemical characterisation of RING1B-HaloTag protein revealed that it was incorporated into PRC1 complexes in a manner that was indistinguishable from the wild type protein (Fig. 1c and Supplementary Fig. 1b, c). When a HaloTag com-patible dye ($JF_{549}$)[66] was applied to RING1B-HaloTag cells, this allowed us to specifically label and image RING1B in the nuclei of live cells, and we observed a nuclear distribution of RING1B signal that was similar to previous reports (Fig. 1d)[24,55,56].

To measure the diffusion and chromatin binding of PRC1, we labelled RING1B-HaloTag with a photoactivatable dye (PA-Halo-$JF_{549}$)[67] and performed high temporal resolution (15 ms expo-sure) SPT with highly inclined and laminated optical sheet (HILO) microscopy[68] (Fig. 2a, b, and Supplementary Movie 1). When the resulting tracks were analysed with Spot-On[69], a three-state kinetic model with immobile, slowly diffusing, and fast diffusing molecules fit the data well. In contrast, a two-state model with only immobile and diffusing molecules fit the data poorly (Supplementary Fig. 2a). These observations suggest there is an immobile fraction of PRC1 which is bound to chromatin, a slowly diffusing fraction that may correspond to transient interactions with chromatin or confinement within areas of the

nucleus, and a fraction that is freely diffusing[70]. Fitting our SPT measurements to this model revealed that 20% of RING1B was chromatin-bound, with the remainder existing in a slowly or freely diffusing state (Fig. 2c). To contextualise these measure-ments, we generated cell lines stably expressing histone H2B-HaloTag, which is stably incorporated into chromatin, or HaloTag with a nuclear localisation signal (HaloTag-3xNLS), which does not bind specifically to chromatin. SPT measurements using these cells yielded different behaviours for each protein that were consistent with previous observations[69], and with their recovery in spot FRAP measurements (Fig. 2c and Supplementary Fig. 2b, c, and Supplementary Movies 2 and 3). H2B-HaloTag had a bound fraction of 59% whereas HaloTag-NLS had a bound fraction of 11%, revealing that the bound fraction of RING1B was towards the lower end of these two extremes (Fig. 2c). Therefore, our SPT measurements reveal that RING1B, and therefore PRC1, exists predominantly in diffusing states and has only a small chromatin-bound fraction.

**Only a fraction of chromatin-bound PRC1 displays stable binding.** High temporal resolution SPT demonstrated that 20% of RING1B was bound to chromatin. However, these imaging conditions did not allow us to measure the length of more stable

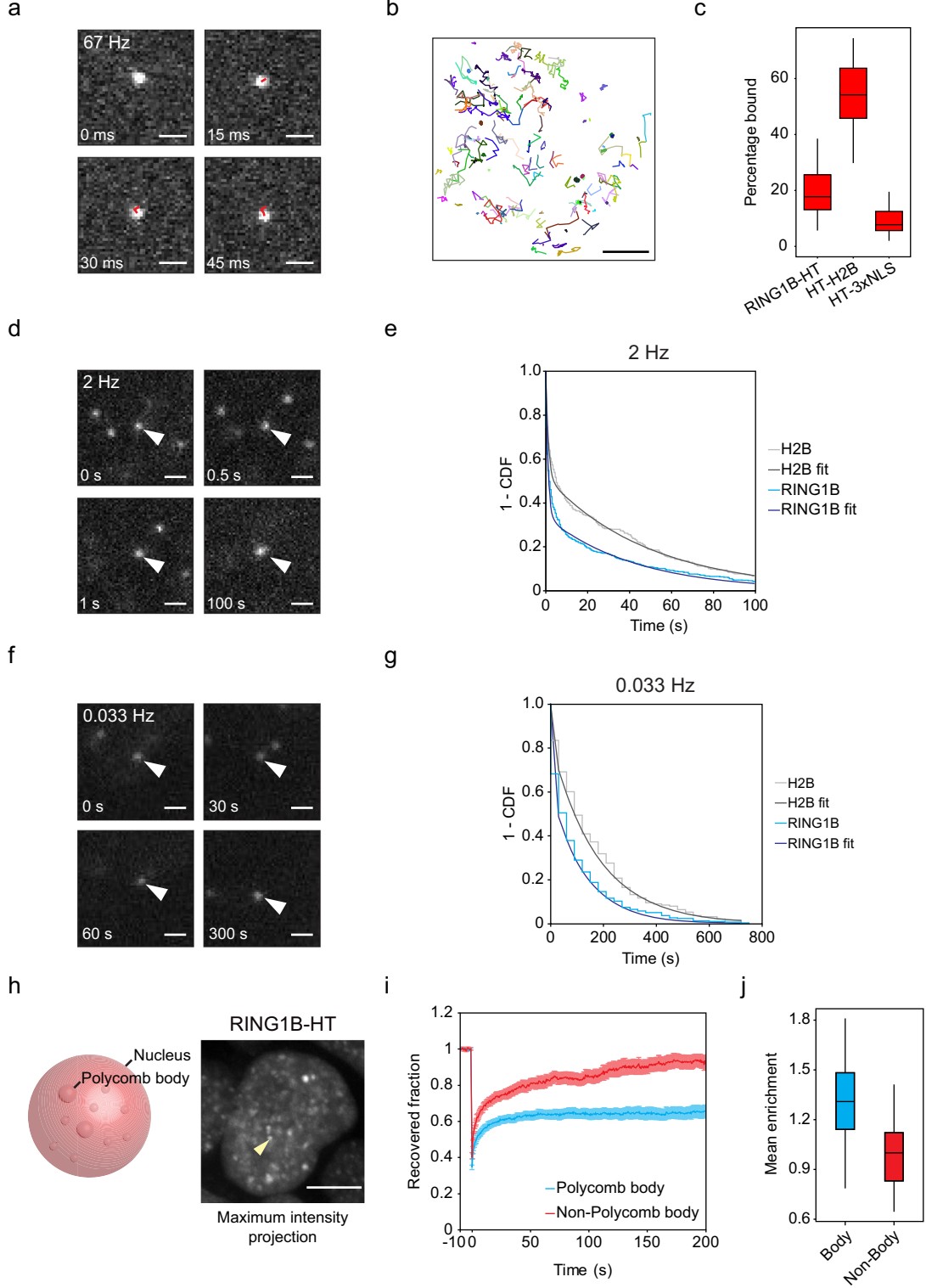

chromatin binding events because individual molecules photobleach within a few seconds. To overcome this limitation and measure the survival times of bound molecules, we carried out 2 Hz SPT using reduced laser power with longer exposure times (0.5 s). Under these imaging conditions, diffusing molecules blur while stably bound molecules are clearly visible and photobleach less rapidly (Fig. 2d and Supplementary Movie 4). The acquisition of these movies was performed with limited photoactivation of molecules such that only a small number of molecules were visible at once. This helped to reduce the probability of tracks of

multiple molecules overlapping and being conflated in our analysis. Using this approach, the distribution of track lengths for chromatin-bound RING1B molecules fit well to a biexponential decay modelling short and long-lived binding events (Fig. 2e). The mean survival times obtained from this model are influenced by photobleaching. Therefore we used H2B-HaloTag, which binds stably to chromatin on the order of hours[71], as a control to correct for the influence of photobleaching[72–75]. However, we were still unable to accurately estimate the duration of long binding events as the observed dwell times for RING1B were

**Fig. 2 PRC1 is highly dynamic with a small stably chromatin-bound fraction. a** Example cropped frames from a 67 Hz exposure SPT movie ($n = 55$) showing a single RING1B molecule. Tracked steps of the molecule between frames are superimposed in red. Scale bar = 1 μm. **b** Example tracks of individual molecules from a single 15 ms exposure RING1B SPT movie out of 55 analysed, showing a range of diffusive behaviours. Scale bar = 3 μm. **c** A box plot indicating the percentage of RING1B, H2B and HaloTag-NLS molecules in the bound state determined from SPT using Spot-On. Box plots represent measurements from $n > 50$ movies each from 2 experiments. Source data are provided as a Source Data file. **d** Example frames from a 2 Hz exposure SPT movie ($n = 24$) showing a single molecule visible for 100 s indicated by white arrowheads. Scale bar = 1 μm. **e** Dwell time distributions (1—cumulative distribution function, CDF) and fitted biexponential decay curves for immobile RING1B molecules for a representative set of movies from a single experiment acquired as in **d**. H2B is included as a photobleaching control for highly stable binding. $n = 24$ movies across 3 experiments. Source data are provided as a Source Data file. **f** Example frames from a 0.033 Hz exposure SPT movie ($n = 10$) showing a single molecule visible for 300 s indicated by white arrowheads. Scale bar = 1 μm. **g** Dwell time distributions (1—cumulative distribution function, CDF) and fitted biexponential decay curves for immobile RING1B molecules for a representative set of movies from a single experiment acquired as in **f**. H2B is included as a photobleaching control for highly stable binding. $n = 10$ movies across 2 experiments. Source data are provided as a Source Data file. **h** Schematic of Polycomb bodies in the nucleus (left panel). Representative maximum intensity projection of nuclear RING1B-HT-JF$_{549}$ signal acquired by spinning disk microscopy ($n = 63$ cells). The yellow arrowhead indicates a single Polycomb body. Scale bar = 5 μm (right panel). **i** FRAP recovery curves for RING1B-HaloTag in regions containing a Polycomb body (blue) or elsewhere in the nucleus (Non-Polycomb body, red). The recovered fraction was measured relative to initial fluorescence intensity and corrected using an unbleached region. The error bars denote SEM for $n = 20$ cells each for Polycomb body and Non-Polycomb body regions across 2 experiments. Source data are provided as a Source Data file. **j** A box plot illustrating the relative mean fluorescence signal per unit volume of all RING1B Polycomb bodies in each cell (Body, blue) compared to the remaining nuclear volume in the same cell (Non-Body, red), normalised to the median non-Polycomb body fluorescence. $n = 63$ cells across 2 experiments. Source data are provided as a Source Data file. **b, j** Boxes represent the interquartile range (IQR), the middle line corresponds to the median, and whiskers extend to the largest and smallest values no more than 1.5 x IQR from the box. Values outside of this range are not plotted, but are included in all analyses.

similar to the photobleaching time (Fig. 2e). Despite this limitation, based on the biexponential decay fit to the distribution of track lengths, we found that 38% of observed binding events were long-lived (stable), whereas 62% were more transient. Additionally, some molecules remained stably bound to chromatin for up to, and beyond, 100 s. To extend the time for which we could image long binding events, we reduced the acquisition frequency to 30 s intervals. Under these conditions, we observed some RING1B molecules that survived for hundreds of seconds, but photobleaching still limited our capacity to accurately estimate stable binding times (Fig. 2f, g, and Supplementary Movie 5). These observations demonstrate that while only 20% of PRC1 is bound to chromatin, and less than half of these binding events are stable, those molecules that bind stably do so for relatively long periods of time.

To investigate whether the stable binding events we observe in SPT are representative of PRC1 behaviour at known sites of chromatin interaction, we focussed on Polycomb bodies. Polycomb bodies are cytologically distinct foci that correspond to a subset of PRC1-bound regions of the genome, including the Hox gene clusters, that have very large Polycomb chromatin domains[24,56,76–78]. Live-cell imaging of RING1B revealed approximately one hundred Polycomb bodies per nucleus (Fig. 2h). Using FRAP measurements, we examined RING1B dynamics and chromatin binding in Polycomb bodies or equally sized regions of the nucleoplasm. While it is challenging to specifically quantify recovery in Polycomb bodies due to their small size and mobility, we nevertheless observed substantially slower and less complete recovery of fluorescence signal within these regions (Fig. 2i). This demonstrates that Polycomb bodies, where PRC1 is known to interact with chromatin, also display stable RING1B binding. However, our measurements also revealed that Polycomb bodies accounted for just 1.3% of the total nuclear volume and that RING1B fluorescence signal inside Polycomb bodies was only 1.3-fold more intense than the surrounding nucleus (Fig. 2j). This means that only 1.7% of total PRC1 signal originates from Polycomb bodies, consistent with evidence from ChIP-seq experiments that PRC1 also binds more stably to thousands of other smaller target sites in the genome[35,36,79,80]. Therefore, PRC1 binds chromatin more stably at Polycomb target sites and our SPT approach allows us to capture these events in live cells throughout the genome.

**Absolute PRC1 quantification estimates low occupancy at PRC1 target sites.** There is currently very limited quantitative understanding of PRC1 abundance and occupancy at target sites in the genome. However, the biochemical makeup of Polycomb chromatin domains will have important implications for how the Polycomb system functions to repress gene expression. Having quantified the fraction of RING1B that is bound to chromatin in live cells, we wanted to use this information to estimate the maximum number of molecules of PRC1 that might bind to a typical target site. To achieve this, we first quantified the number of RING1B molecules in ESCs using a biochemical approach[81] that compares in-gel fluorescence of RING1B-HaloTag from a defined number of cells to the fluorescence of a recombinant HaloTag protein of known concentration (Fig. 3a). From this we calculated that there were ~63,000 molecules of RING1B per cell, which is similar to estimates in other cell types[82–84]. SPT demonstrated that only 20% of RING1B was bound to chromatin (Fig. 2c), which corresponds to ~12,600 bound molecules. To estimate the maximum number of molecules that could possibly bind to RING1B-occupied target sites identified in ChIP-seq (18,643 sites)[79], we assumed that the bound fraction of RING1B was concentrated exclusively in these sites. Based on this liberal assumption, we estimate that there would be on average 0.1 RING1B molecules for every kilobase of RING1B-enriched chromatin. However, the distribution of RING1B ChIP-seq signal varies significantly across its binding sites in the genome (Fig. 3b, c). Even if this distribution is taken into consideration, sites in the top decile of RING1B density would still have fewer than 0.3 molecules per kb. Despite being very small, these occupancy values will be an overestimation because we know that PRC1 binding events also occur away from sites identified in RING1B ChIP-seq as evidenced by pervasive H2AK119ub1 throughout the genome[49,85]. Therefore, our calculations suggest that PRC1 occupancy at target sites in the genome is on average very low.

**Interaction between the PRC1 catalytic core and the nucleosome is not a central determinant of chromatin binding.** Having characterised chromatin binding by PRC1 in SPT, we then set out to dissect the mechanisms that define this behaviour. RING1B binds to the nucleosome in a specific orientation and

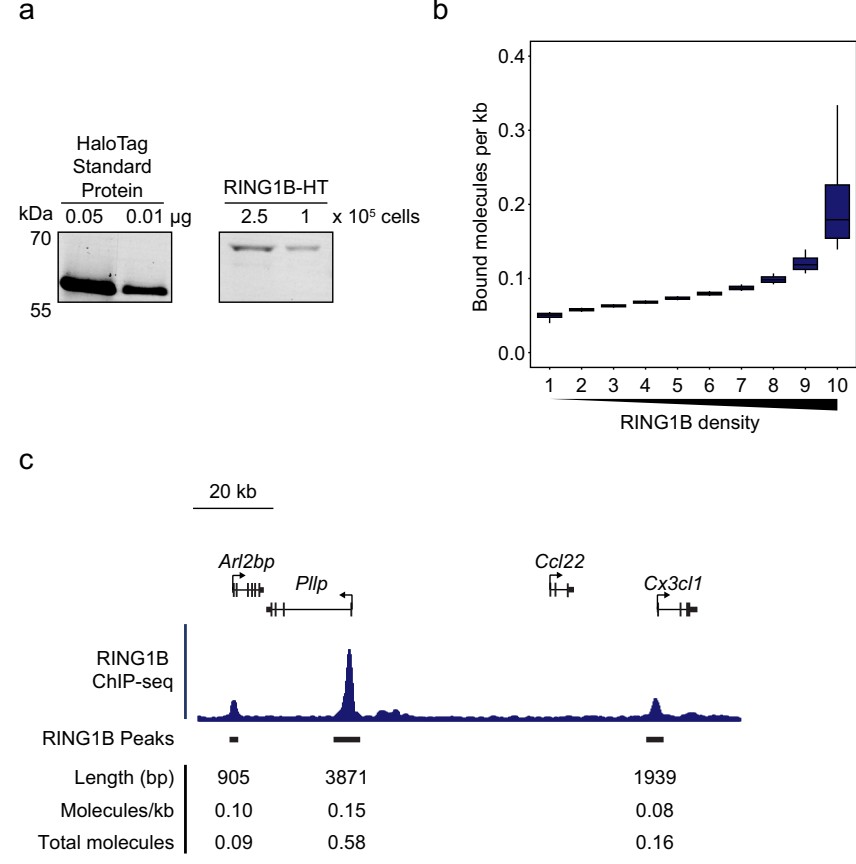

**Fig. 3 RING1B has low target site occupancy. a** An example of an in-gel fluorescence image of TMR-labelled HaloTag Standard Protein loaded in known quantities (left panel) and RING1B-HT-TMR from whole-cell lysates of a known number of cells (right panel). Fluorescence intensity from $n = 3$ biological replicates was used to quantify the number of molecules present[81]. Source data are provided as a Source Data file. **b** A box plot illustrating the number of RING1B molecules that are estimated to be bound per kilobase of DNA at RING1B peaks from ChIP-seq in ESCs[79]. Read distributions aggregated from $n = 3$ biological replicates of RING1B ChIP-seq were used to segregate bound RING1B molecules into density deciles over RING1B peaks (18,643). Boxes represent the interquartile range (IQR), the middle line corresponds to the median, and whiskers extend to the largest and smallest values no more than 1.5 x IQR from the box. Values outside of this range are not plotted, but are included in all analyses. Source data are provided as a Source Data file. **c** A genomic snapshot of RING1B ChIP-seq illustrating the varying distribution of RING1B binding between individual peaks. The values shown below indicate the length of the RING1B peak, the estimated density of RING1B molecules bound at each peak, and the estimated number of molecules bound per allele to each peak assuming a 2n genome.

through a conserved set of residues to ubiquitylate H2A. Mutating these residues renders RING1B incapable of binding to nucleosomes and catalysing H2AK119ub1 in vitro[86,87]. Therefore, we reasoned that nucleosome interaction might contribute centrally to chromatin binding in vivo. To test this, we substituted key residues in the nucleosome-binding domain (RING1B$^{NBM}$) and stably expressed RING1B$^{NBM}$-HaloTag or wild type RING1B-HaloTag in a cell line that lacks RING1A and where removal of endogenous RING1B can be induced by addition of tamoxifen (PRC1$^{CKO}$) (Fig. 4a and b)[79,86,87]. Following tamoxifen treatment, cells expressing wild type RING1B-HaloTag retained H2AK119ub1, had normal cell morphology, and remained pluripotent as measured by alkaline phosphatase staining of the resulting colonies (Fig. 4b and Supplementary Fig. 3a). Importantly, RING1B$^{NBM}$ failed to maintain these features, demonstrating it was non-functional in vivo and caused a loss of normal ESC viability and pluripotency. To investigate the effects that mutating the nucleosome-binding domain had on RING1B binding we carried out 67 Hz and 2 Hz SPT for RING1B$^{NBM}$. The loss of H2AK119ub1 and viability in cell lines expressing only the RING1B$^{NBM}$ necessitated that we image this mutant in the presence of endogenous RING1B to avoid indirect effects. Although we cannot rule out that endogenous RING1B might influence the dynamics of the RINGB$^{NBM}$, we found that the

fraction of bound and stably bound molecules for RING1B$^{NBM}$ was almost identical to that of the wild type protein (Fig. 4c, d) and stable binding remained beyond the photobleaching limit of 2 Hz imaging. This suggests that the interaction between RING1B and the nucleosome does not contribute centrally to chromatin binding by PRC1. In agreement with these findings, a minimal catalytic domain that does not interact with PRC1 auxiliary factors, but is capable of depositing H2AK119ub1, had a chromatin bound fraction (8%) that was indistinguishable from the HaloTag-NLS control (Supplementary Fig. 3b and Fig. 4e, f)[86,88–90]. Therefore, we conclude that interaction between the catalytic core of PRC1 and the nucleosome does not contribute centrally to chromatin binding, and propose instead that this must rely on auxiliary proteins in PRC1 complexes.

**PCGF2-PRC1 is the most abundant PRC1 complex in ESCs.** PCGF proteins dimerise with RING1B and interact with auxiliary PRC1 subunits that are thought to contribute to chromatin binding by PRC1[31,32,34,35]. Therefore, we reasoned that understanding the composition of PRC1 complexes and defining their individual chromatin binding behaviours in live cells would be required to discover the mechanisms that

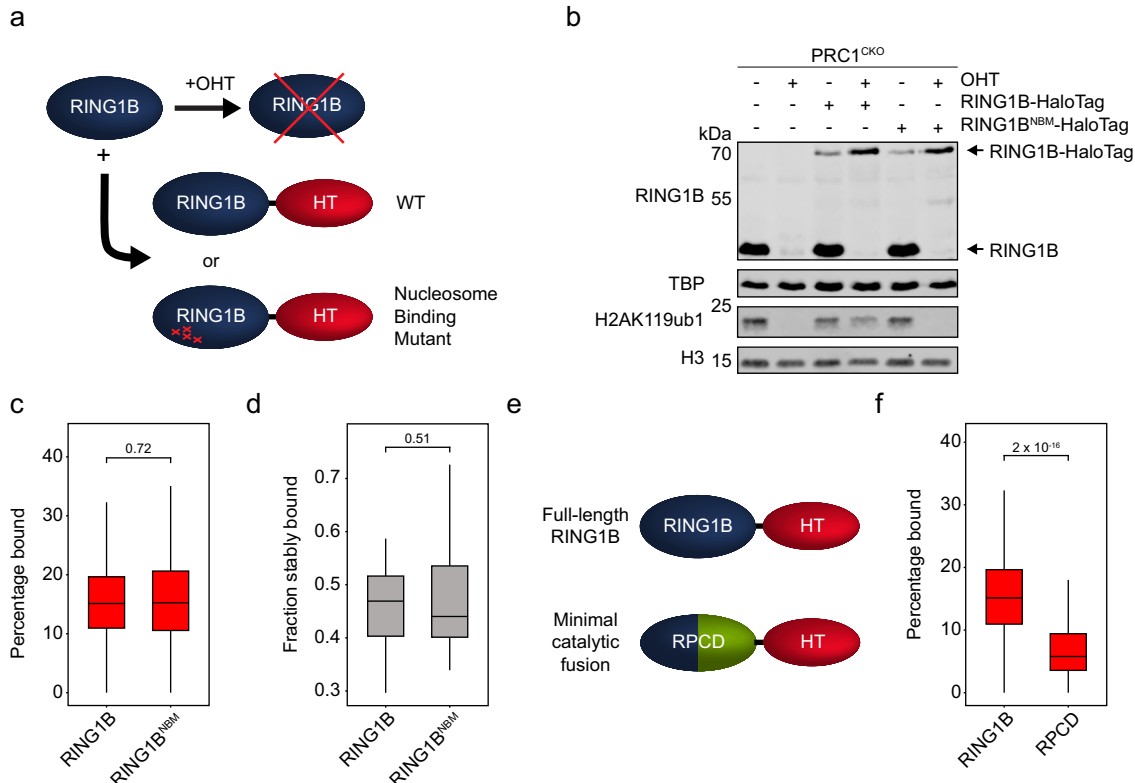

**Fig. 4 Interaction between the PRC1 catalytic core and the nucleosome is not a central determinant of chromatin binding. a** A schematic illustrating the cell line used for exogenous expression of RING1B-HaloTag fusions. Either wild type (WT) RING1B or a nucleosome binding mutant (NBM) fused to the HaloTag was stably expressed in *Ring1a$^{-/-}$; Ring1b$^{fl/fl}$* (PRC1$^{CKO}$) cells. Addition of OHT causes removal of endogenous RING1B. **b** Western blots showing loss of endogenous RING1B and H2AK119ub1 following addition of OHT in PRC1$^{CKO}$ cells and rescue of H2AK119ub1 by RING1B-HaloTag but not RING1B$^{NBM}$-HaloTag. TBP and H3 are included as loading controls. $n = 1$ biological replicate. Source data are provided as a Source Data file. **c** A box plot indicating the percentage of RING1B and RING1B$^{NBM}$ molecules in the bound state as determined by SPT. Box plots represent measurements from $n > 90$ movies of different cells from 3 experiments. The indicated *p*-value was calculated using a two-tailed Student's *t* test. Source data are provided as a Source Data file. **d** A box plot showing the fraction of RING1B and RING1B$^{NBM}$ molecules that exhibit stable binding. Box plots represent measurements from $n = 16$ movies from 2 experiments. The indicated *p*-value was calculated using a two-tailed Student's *t* test. **e** Schematic showing the minimal RING1B PCGF4 catalytic domain (RPCD) formed by fusion of the RING domains of RING1B and PCGF4. Source data are provided as a Source Data file. **f** A box plot of percentages of RING1B and RPCD molecules in the bound state. Box plots represent measurements from $n > 90$ movies from 3 experiments. The indicated *p* value was calculated using a two-tailed Student's *t* test. Source data are provided as a Source Data file. In **c**, **d**, **f** boxes represent the interquartile range (IQR), the middle line corresponds to the median, and whiskers extend to the largest and smallest values no more than 1.5 x IQR from the box. Values outside of this range are not plotted, but are included in all analyses.

underpin PRC1 targeting and function in gene repression. To achieve this, we added a HaloTag to both alleles of the endogenous *Pcgf1*, *Pcgf2*, *Pcgf3*, and *Pcgf6* genes (Fig. 5a, b, and Supplementary Fig. 4a, c, d, e, g, h). PCGF4 is not expressed in ESCs so we excluded it from our analysis. PCGF3 and 5 form nearly identical PRC1 complexes and are expressed at similar levels, so PCGF3 was used as a proxy for PCGF3/5-PRC1 complex behaviour[31,32,35]. In ESCs, canonical PRC1 complexes form around PCGF2 and predominantly contain CBX7, whereas variant PRC1 complexes form mostly around PCGF1, 3, 5, and 6 and predominantly contain RYBP (Fig. 5a)[31,32,35,36,39]. Therefore, to capture chromatin binding by canonical and variant PRC1 complexes, we added a Halo-Tag to both alleles of the *Cbx7* and *Rybp* genes (Fig. 5b and Supplementary Fig. 4b, f).

To determine the abundance of individual PRC1 complexes in ESCs we first quantified the number of PCGF molecules and in parallel determined their relative levels by fluorescence microscopy. This showed that individual PCGF molecules were expressed at lower levels than RING1B, consistent with each PCGF complex constituting a fraction of total PRC1 (Fig. 5c).

PCGF2 was most abundant with ~26,000 molecules per cell and corresponded to 40% of total RING1B. The number of CBX7 molecules (~19,000) was similar to that of PCGF2, suggesting that canonical PCGF2/CBX7-PRC1 complexes are the predominant form of PRC1 in ESCs. Differences in the absolute number of PCGF2 and CBX7 molecules likely result from some PCGF2 forming variant complexes[31,36,39] and the incorporation of other more lowly expressed CBX proteins in PCGF2-PRC1 complexes[32,35]. In contrast, PCGF proteins that exclusively form variant PRC1 complexes were less abundant. There were ~14,000 PCGF6, ~5000 PCGF1, and ~3000 PCGF3 molecules per cell. Given that PCGF3 and PCGF5 are expressed at similar levels (Supplementary Fig. 4d)[35,49,91], we estimate that there are ~6000 PCGF3/5 molecules per cell. At ~30,000 molecules per cell, RYBP was slightly more abundant than PCGF1, 3, 5, and 6 together, in agreement with a small amount of PCGF2 forming variant PRC1 complexes[31,32,36,39]. Importantly, the total sum of PCGF molecules, or CBX7/RYBP molecules, was similar to the number of RING1B molecules, consistent with RING1B forming a core scaffold for PRC1 complex assembly[11,22,49,65,89,92,93]. This detailed quantification of canonical and variant PRC1 subunit

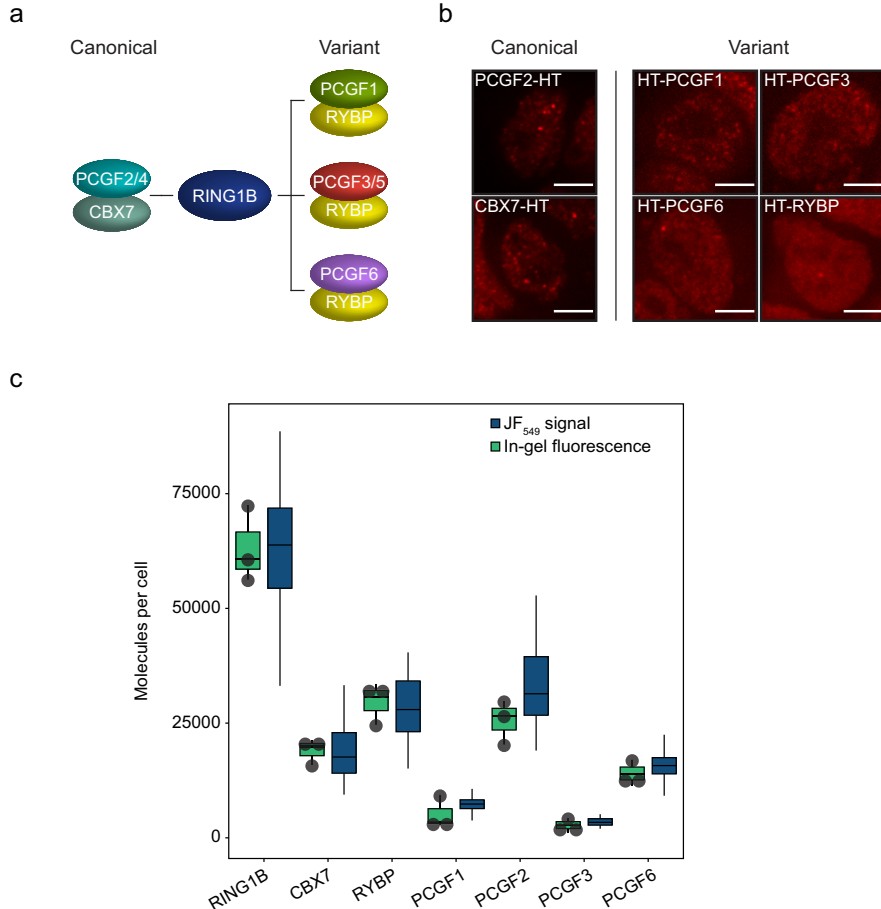

**Fig. 5 PCGF2-PRC1 is the most abundant PRC1 complex in ESC. a** A schematic illustrating the central canonical and variant PRC1 complexes.
**b** Representative images showing a single Z-slice from ESCs expressing HaloTag fusions of PRC1 subunits labelled with $JF_{549}$. Scale bar = 5 μm. Imaging was performed in $n = 2$ biological replicates. **c** A box plot showing estimates of the number of protein molecules per cell, measured by in-gel fluorescence (green), and corresponding measurements of relative $JF_{549}$ signal for PRC1 subunits (blue). Measured fluorescence intensities have been converted into numbers of molecules by normalisation to the mean number of RING1B molecules per cell. Box plots indicate measurements for $n > 35$ cells ($JF_{549}$ signal) or $n = 3$ independent biological replicates of in-gel quantifications (In-gel fluorescence). The means for each replicate of in-gel fluorescence measurements are represented by black dots. Boxes represent the interquartile range (IQR), the middle line corresponds to the median, and whiskers extend to the largest and smallest values no more than 1.5 x IQR from the box. Values outside of this range are not plotted, but are included in all analyses. Source data are provided as a Source Data file.

abundance suggests that canonical PRC1 is the most abundant form of PRC1 in ESCs.

**Canonical PRC1 is characterised by stable chromatin binding and is restricted to a subset of PRC1 bound sites.** Based on our quantification of PRC1 complex abundances, we then set out to discover how these complexes enable the chromatin binding behaviour and function of PRC1. Initially we focussed on the canonical PRC1 components PCGF2/CBX7 and carried out SPT. This revealed that 18% of PCGF2 and CBX7 were chromatin bound, which was similar to RING1B (20%) (Fig. 6a). 2 Hz survival time imaging revealed that PCGF2 and CBX7 had a slightly larger fraction of stably bound molecules (44 and 50%) than RING1B (38%) with their stable binding time estimations also being limited by photobleaching (Fig. 6b). It has been shown that canonical PRC1 is important for the formation of Polycomb bodies[24,94,95]. Consistent with this, when we examined the distribution of PCGF2 and CBX7 in the nucleus, both were more enriched (approximately 2.3-fold and 1.7-fold respectively) in

Polycomb bodies than RING1B (Supplementary Fig. 5a). Furthermore, FRAP for PCGF2 and CBX7 in Polycomb bodies revealed delayed and incomplete recovery relative to regions of the nucleus without Polycomb bodies (Supplementary Fig. 5b). This suggests that, similar to RING1B, there is more stably bound PCGF2 and CBX7 within Polycomb bodies. We estimate that on average each Polycomb body contains 9 PCGF2 and 10 RING1B molecules, suggesting that these foci are mostly composed of canonical PRC1 complexes and is consistent with the more prominent enrichment of PCGF2/CBX7 in these foci compared to variant PRC1 subunits (Fig. 5b). However, despite the visible enrichment of PCGF2 and RING1B in Polycomb bodies, we estimate that there are relatively few molecules bound at these sites, as even the brightest foci correspond to not more than ~50 PRC1 complexes, and these regions represent only a small fraction of total PRC1 (1.7% for RING1B, 3.6% for PCGF2 and 3.4% for CBX7). Together, these observations suggest that while canonical PRC1 is more restricted to Polycomb bodies than PRC1 as a whole, Polycomb bodies in ESCs do not correspond to large accumulations of Polycomb proteins.

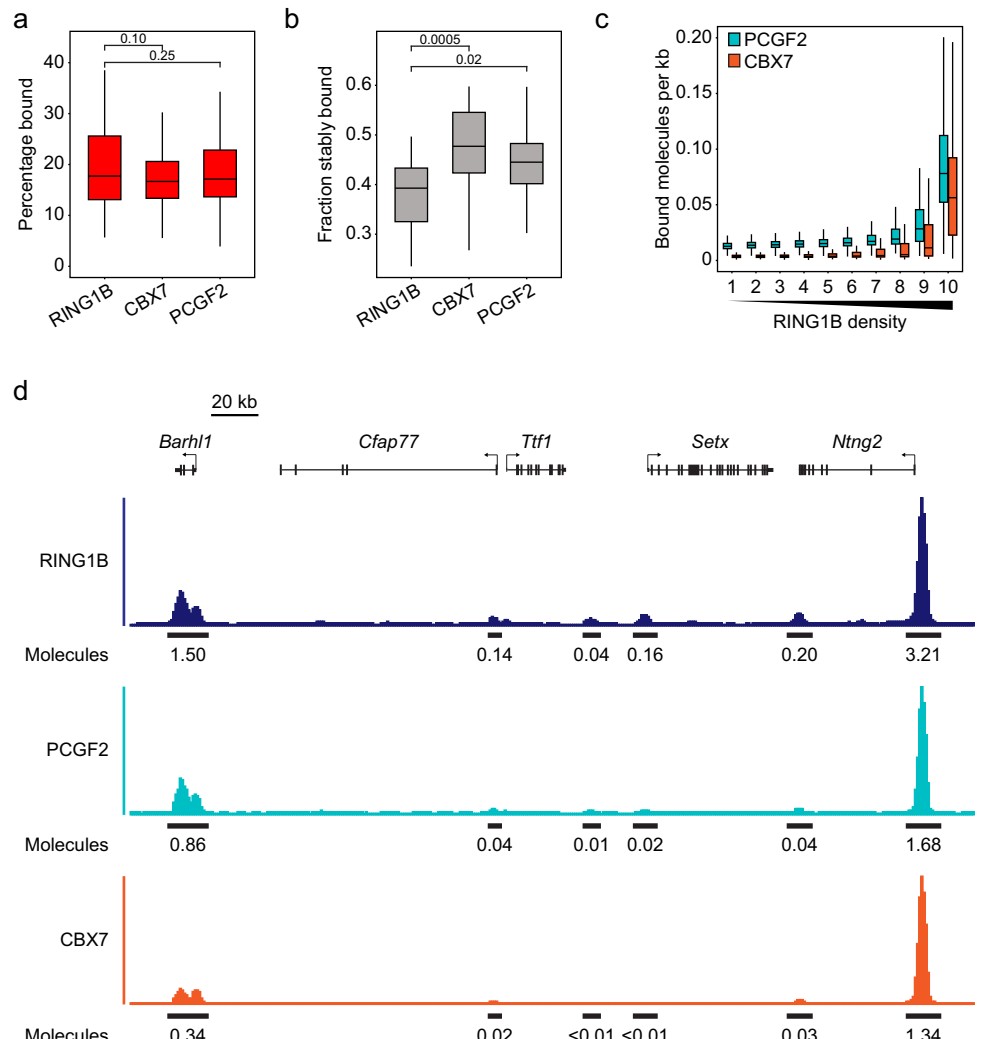

**Fig. 6 Canonical PRC1 exhibits stable chromatin binding and is restricted to a subset of Polycomb sites. a** A box plot of the percentage of RING1B, CBX7 and PCGF2 molecules in the bound state as determined by SPT. Box plots represent measurements from $n > 50$ movies from 3 experiments each. Indicated $p$-values were calculated using a two-tailed Student's $t$ test. Source data are provided as a Source Data file. **b** A box plot indicating the fraction of RING1B, CBX7 and PCGF2 molecules exhibiting stable binding. Box plots represent measurements from $n > 15$ movies from at least 2 experiments. Indicated $p$-values were calculated using a two-tailed Student's $t$ test. Source data are provided as a Source Data file. **c** A box plot illustrating the number of PCGF2 (turquoise) and CBX7 (orange) molecules that are estimated to be bound per kilobase of DNA at RING1B peaks from ChIP-seq in ESCs[49,79]. Read distributions aggregated from $n = 3$ biological replicates of PCGF2 and CBX7 ChIP-seq were used to segregate bound PCGF2 and CBX7 molecules into density deciles over RING1B peaks. Source data are provided as a Source Data file. **d** A genomic snapshot illustrating RING1B, PCGF2 and CBX7 ChIP-seq signal at two strong PRC1 peaks sites and four weaker peaks illustrating the more restricted nature of PCGF2 and CBX7 binding. Values shown below the ChIP-seq signal indicate the estimated number RING1B, PCGF2 and CBX7 molecules bound at a single allele for the indicated peak. In **a–c**, boxes represent the interquartile range (IQR), the middle line corresponds to the median, and whiskers extend to the largest and smallest values no more than 1.5 x IQR from the box. Values outside of this range are not plotted, but are included in all analyses.

Previous ChIP-seq analysis has shown that the distributions of PCGF2 and RING1B are not identical[35,36,49,79,91]. By combining information detailing PCGF2 and CBX7 binding from ChIP-seq analysis and our estimates of the number of bound molecules in live-cell imaging, we found that both PCGF2 and CBX7 occupancy was most dense at Polycomb target sites in the top decile of RING1B density (0.09 and 0.07 molecules per kb). Occupancy then decayed quickly (to between 0.01 and 0.03 molecules per kb) at sites with lower levels of RING1B, with CBX7 occupancy being almost negligible outside of the highest two deciles of RING1B occupancy (Fig. 6c, d). PRC2 is also most enriched at high density RING1B sites in ChIP-seq and present in Polycomb bodies[79] which have elevated levels of H3K27me3. This is consistent with the restricted nature of PCGF2 occupancy

relying on CBX proteins in canonical PRC1 complexes, including CBX7, that can bind to H3K27me3[35,36,39].

**Variant PRC1 complexes are characterised by distinct dynamics and chromatin binding.** Having demonstrated that canonical PRC1 displays stable chromatin binding and occupancy at a restricted subset of PRC1 bound sites, we then set out to examine the behaviour of variant PRC1 complexes and to determine their contribution to PRC1 complex dynamics and chromatin binding. Interestingly, SPT revealed that variant PRC1 complexes had dynamics that were distinct from canonical PRC1. PCGF3 and PCGF6 had smaller bound fractions than RING1B (13% and 17% respectively) and were more similar to that of

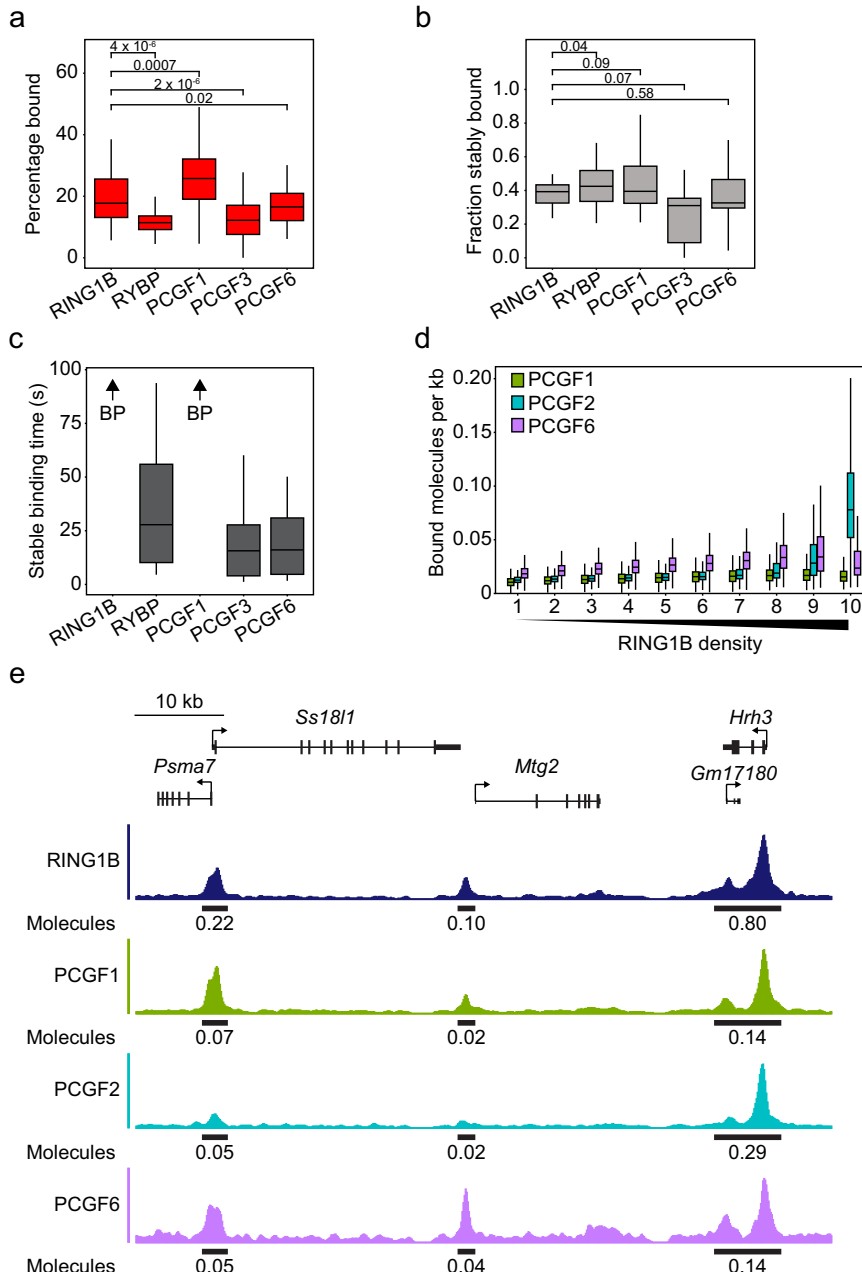

**Fig. 7 Variant PRC1 subunits exhibit distinct dynamics and broad binding to PRC1 targets. a** A box plot indicating the percentage of RING1B, RYBP, PCGF1, PCGF3 and PCGF6 molecules in the bound state determined by SPT. Box plots represent measurements from $n > 50$ movies from 3 experiments each. Indicated $p$-values were calculated using a two-tailed Student's $t$ test. Source data are provided as a Source Data file. **b** A box plot indicating the fraction of RING1B, RYBP, PCGF1, PCGF3 and PCGF6 molecules that exhibit stable binding. Box plots represent measurements from $n > 15$ movies from at least 2 experiments each. Indicated $p$-values were calculated using a two-tailed Student's $t$ test. Source data are provided as a Source Data file. **c** A box plot indicating the stable binding times for RING1B, RYBP, PCGF1, PCGF3 and PCGF6 after photobleaching correction in 2 Hz tracking experiments. BP (beyond photobleaching) and an arrow indicates proteins for which stable binding exceeded the photobleaching time. Box plots represent measurements from $n > 15$ movies from at least 2 experiments each. Source data are provided as a Source Data file. **d** A box plot illustrating the number of PCGF1 (green), PCGF2 (turquoise), PCGF6 (purple) molecules that are estimated to be bound per kilobase of DNA at RING1B peaks from ChIP-seq in ESCs[79]. Read distributions aggregated from $n = 3$ biological replicates of corresponding ChIP-seq were used to segregate bound molecules into density deciles over RING1B peaks. Source data are provided as a Source Data file. **e** A genomic snapshot of RING1B, PCGF1, PCGF2 and PCGF6 ChIP-seq illustrating the differing distributions of PRC1 complexes across RING1B peaks. Values shown below the ChIP-seq signal indicate the estimated number of RING1B, PCGF1, PCGF2 and PCGF6 molecules bound at a single allele for the indicated peak. In **a–d**, boxes represent the interquartile range (IQR), the middle line corresponds to the median, and whiskers extend to the largest and smallest values no more than 1.5 x IQR from the box. Values outside of this range are not plotted, but are included in all analyses.

RYBP (13%) (Fig. 7a). In contrast, PCGF1 had the highest bound fraction (26%) of any PCGF protein. 2 Hz survival time imaging revealed that, although all variant PRC1 subunits had similar stably bound fractions to RING1B, the average length of stable

binding events was between 30 and 50 seconds for RYBP, PCGF3, and PCGF6 (Fig. 7b, c). This is much shorter than the stable binding events observed for canonical PRC1, which extended beyond the photobleaching limit of our experiments. In contrast,

PCGF1 displayed long binding times that were indistinguishable from canonical PRC1. This indicates that the PCGF1-PRC1 complex binds to chromatin more frequently and stably than other variant PRC1 complexes. When we imaged the distribution of PCGF1, PCGF3 and RYBP in the nucleus, these variant PRC1 complex components were not sufficiently enriched in Polycomb bodies to segment these regions from the rest of the nucleus. Some PCGF6 was evident in Polycomb bodies, but importantly this was far less enriched (1.2-fold) than PCGF2 and CBX7 (2.3-fold and 1.7-fold respectively), and we estimate that there are on average only 1–2 molecules of PCGF6 per Polycomb body (Supplementary Fig. 6). These findings are consistent with our observation that PCGF2 appears to account for the majority of PRC1 in Polycomb bodies. Furthermore, it indicates that variant PRC1 complexes are not central components of Polycomb bodies and likely function more broadly in the genome.

The differences between the dynamics of variant PRC1 complexes, we observed in SPT suggest they must have distinct chromatin binding mechanisms. We reasoned that contextualising these behaviours was important as variant PRC1 complexes are responsible for almost all H2AK119ub1 in the genome and have recently been proposed to be central to PRC1-mediated gene repression[49,91]. We first considered PCGF3 dynamics in the context of its known role in depositing low levels of H2AK119ub1 indiscriminately across the genome[49]. Our attempts to ChIP-seq PCGF3/5 have yielded no obvious enrichment profile, with similar approaches yielding very few enriched sites[91]. An inability to capture PCGF3/5-PRC1 by ChIP is consistent with our SPT measurements showing that PCGF3 has a small chromatin-bound fraction and less stable binding than other PCGF molecules. Furthermore, this is consistent with previous reports suggesting that fixation based chromatin binding assays fail to effectively capture transient chromatin binding events[96,97]. More importantly however, we posit that the dynamic chromatin interactions that PCGF3-PRC1 engages in may be ideally suited to the unique role PCGF3/5 play in depositing H2AK119ub1 pervasively throughout the genome[49]. In line with this, we estimate that there are $5.9 \times 10^6$ H2AK119ub1 molecules in a diploid ESC genome and H2AK119ub1 has a half-life of ~90 min[98], necessitating replacement of 750 molecules of H2AK119ub1 by PRC1 each second. Approximately 50% of H2AK119ub1 is deposited by PCGF3/5-PRC1 complexes[49], of which we estimate there are ~6000 molecules. Each PCGF3/5-PRC1 complex would therefore need to carry out on average at least one ubiquitylation event every 16 s. Importantly, the binding site search time[62,75] for the most prominent PCGF3 binding modality (short binding, ~1 s) was ~15 s, consistent with the rate of H2AK119ub1 deposition that would be required to maintain the levels of the modification. These calculations suggest that the highly dynamic nature of PCGF3 relative to other PRC1 subunits may be necessary for the role it plays in depositing H2AK119ub1 throughout the genome.

In contrast to PCGF3/5, ChIP-seq approaches have shown that PCGF1- and PCGF6-PRC1 complexes bind predominantly at PRC1 target genes. The capacity to capture these proteins in ChIP-seq is consistent with their higher bound fraction and longer binding times in SPT experiments. By integrating PCGF1 protein numbers, the bound fraction from SPT, and ChIP-seq distributions, we estimate a narrow range of PCGF1 occupancy (between 0.01 to 0.02 molecules per kb) across PRC1 target sites (Fig. 7d, e). Despite the distribution of PCGF6 ChIP-seq signal correlating slightly better with RING1B at PRC1 target sites, similar estimations for PCGF6 show that even at sites with the highest density of PCGF6, there would on average only be 0.04 molecules of PCGF6 per kb (Fig. 7d, e). Therefore, in contrast to canonical PRC1, variant PCGF1 and PCGF6-PRC1 complexes

have a more monotone distribution across PRC1 target sites and even lower occupancy levels. This suggests that low-level occupancy of variant PRC1 complexes at target sites is sufficient to maintain enrichment of H2AK119ub1 and support gene repression.

**PRC2 and H3K27me3 regulate the frequency of chromatin binding by canonical PRC1.** Our SPT experiments demonstrated that PCGF2- and PCGF1-PRC1 complexes had similar stable chromatin binding. However, these PRC1 complexes have different distributions at target sites in the genome, are proposed to utilise distinct chromatin binding mechanisms, and contribute uniquely to Polycomb system function. Therefore, we were keen to explore the underlying mechanisms that enable these kinetically similar chromatin-binding behaviours but lead to distinct targeting and function. Initially we focussed on the canonical PCGF2-PRC1 complex because its CBX proteins bind H3K27me3 at PRC1 target sites and this modification is known to be important for occupancy as measured by ChIP[16,39,99,100]. To examine how H3K27me3 affects PCGF2 dynamics and chromatin binding in live cells we used the recently developed degradation tag (dTAG) system[101] to induce disruption of the enzyme complex that places H3K27me3. To achieve this, we introduced a dTAG into both alleles of the endogenous *Suz12* gene. SUZ12 is a core structural subunit of PRC2 and is required for its histone methyltransferase activity[102–104]. Treatment of dTAG-SUZ12 cell lines with the dTAG-13 compound caused rapid degradation of SUZ12 and a near complete loss of H3K27me3 by 96 h of treatment (Fig. 8a, b). We used this approach to induce removal of SUZ12 in PCGF2-HaloTag and RING1B-HaloTag cell lines while using a HaloTag-PCGF6 cell line as a control for a PRC1 complex that lacks H3K27me3-binding CBX proteins.

We then carried out SPT prior to and following SUZ12/ H3K27me3 depletion. After dTAG-13 treatment, the bound fraction of PCGF2 and RING1B was modestly yet significantly reduced, in agreement with observations from ChIP experiments that loss of H3K27me3 reduces canonical PRC1 occupancy (Fig. 8c)[16,39,99,100,105]. Importantly, PCGF6-PRC1 chromatin binding was unaffected. Interestingly, despite a reduction in the total bound fraction of RING1B and PCGF2, there was no significant effect on the fraction of stable binding events observed in 2 Hz survival time imaging (Fig. 8d, e). Furthermore, when we examined Polycomb bodies, there was only a modest effect on the number of PCGF2 foci and a slight downwards shift in the distribution of PCGF2 foci volumes, consistent with the small reduction in bound fraction for RING1B and PCGF2 (Fig. 8f, g and h, and Supplementary Fig. 7a, b). This suggests that H3K27me3 recognition is primarily required to increase the frequency with which canonical PRC1 binds chromatin, but not the stability of these binding events when they occur, and that this binding only contributes modestly to canonical PRC1 incorporation into Polycomb bodies. Furthermore, it indicates that H3K27me3-independent mechanisms predominate in supporting stable chromatin binding by canonical PRC1.

**The C-terminal RAWUL domain of PCGF1 is required for stable chromatin binding.** Although PCGF1 exhibited similar binding stability to canonical PRC1 subunits, these PRC1 complexes lack shared auxiliary subunits. This suggested the mechanisms that support stable chromatin binding by PCGF1-PRC1 must be distinct from those of PCGF2-PRC1. Therefore, we were keen to define how PCGF1 interfaces with chromatin. PCGF1 can be separated into two distinct regions based on domain conservation (Supplementary Fig. 8a)[48,106,107]. The N-terminus of PCGF1 contains a RING domain through which it

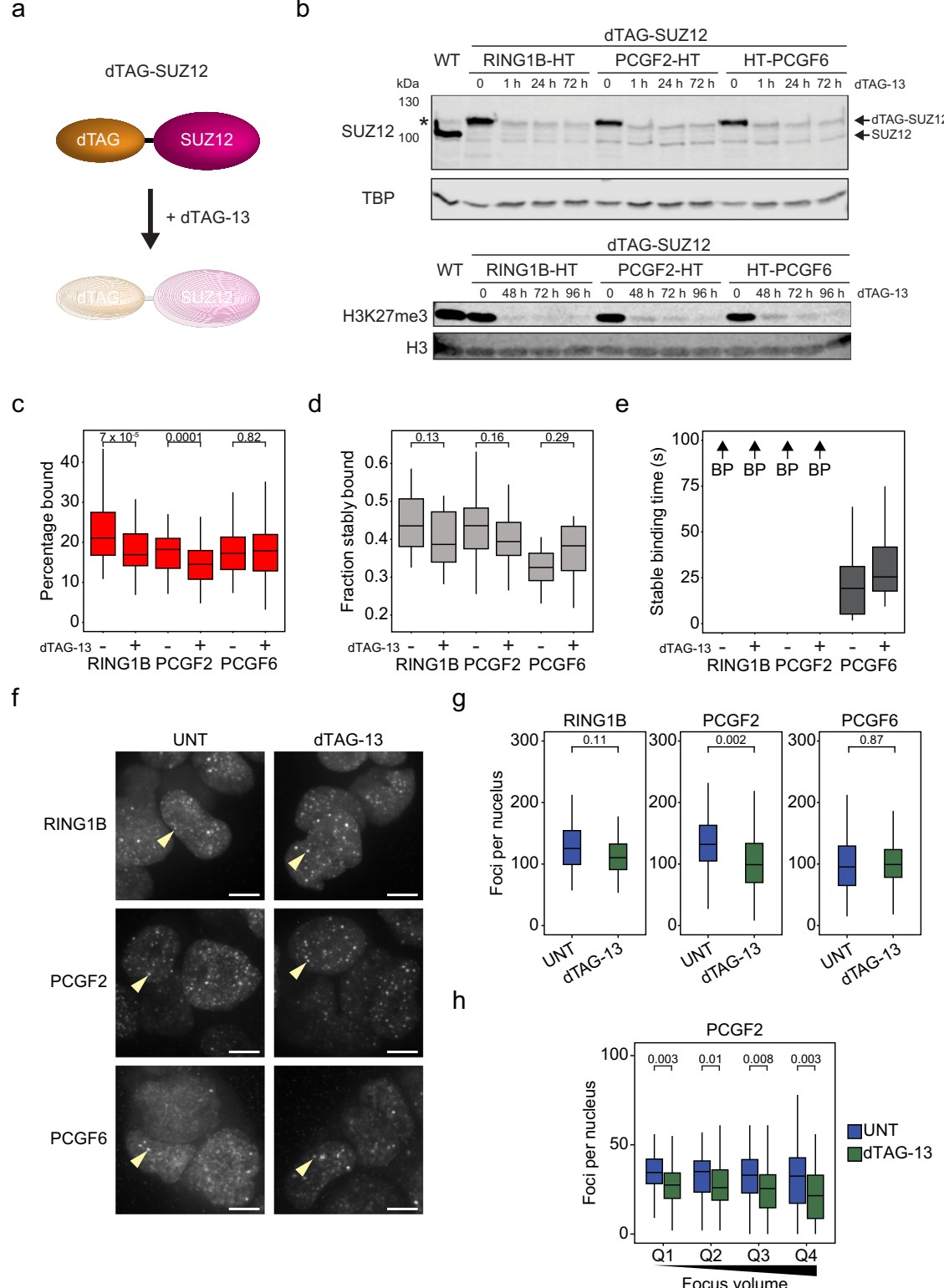

heterodimerises with RING1B, forming a RING-RING domain dimer that is responsible for the ubiquitin ligase activity of the complex. In contrast, the C-terminus of PCGF1 contains a RAWUL domain which interacts with the PCGF1-PRC1 complex specific proteins KDM2B and BCOR/BCORL1[34,38,48,107]. In order to determine which of these domains and their interactions might explain the chromatin binding behaviour of PCGF1 we generated cell lines stably expressing exogenous HaloTag-PCGF1 fusion

proteins corresponding to the full-length protein and its N- and C-terminal domains (Fig. 9a and Supplementary Fig. 8a, b). Full-length PCGF1 exhibited stable binding that was comparable to that of the endogenous protein, although it had a smaller bound fraction (Fig. 9b, c). When we examined the behaviour of the N- and C-terminal fragments, the bound fraction for each was similar to the full-length protein, with a modest reduction in the binding frequency of the C-terminal fragment (Fig. 9b).

**Fig. 8 PRC2 and H3K27me3 contribute to the binding frequency of canonical PRC1 but not its stability. a** A schematic illustrating the dTAG-SUZ12 system. SUZ12 is tagged endogenously and is depleted by addition of dTAG-13. **b** Western blots for RING1B-HT, PCGF2-HT and HT-PCGF6 cell lines with the dTAG-SUZ12 system showing rapid (<1 hr) depletion of SUZ12 and loss of H3K27me3 after 96 hrs of dTAG-13 treatment. TBP and H3 are included as loading controls. Symbol (*) indicates a nonspecific band present in the parental cell line which remains after SUZ12 depletion. $n = 1$ biological replicate. Source data are provided as a Source Data file. **c** A box plot indicating the percentage of RING1B, PCGF2, and PCGF6 molecules in the bound state with and without 96 hr dTAG-13 treatment, as determined from SPT. Box plots represent measurements from $n = 90$ movies from 2 experiments each. Indicated $p$-values were calculated using a two-tailed Student's $t$ test. Source data are provided as a Source Data file. **d** A box plot indicating the fraction of RING1B, PCGF2, and PCGF6 molecules exhibiting stable binding with and without 96 hr dTAG-13 treatment. Box plots represent measurements from $n = 16$ movies from 2 experiments each. Indicated $p$-values were calculated using a two-tailed Student's $t$ test. Source data are provided as a Source Data file. **e** A box plot indicating the stable binding times for RING1B, PCGF2, and PCGF6 with and without 96 hr dTAG-13 treatment. BP (beyond photobleaching) and an arrow indicates proteins for which stable binding exceeded the photobleaching time. Box plots represent measurements from $n = 16$ movies from 2 experiments each. Source data are provided as a Source Data file. **f** Maximum intensity projections of nuclear RING1B, PCGF2 and PCGF6 signal with and without 96 hr dTAG-13 treatment. Yellow arrowheads indicate single Polycomb bodies. Scale bar = 5 μm. $n > 50$ cells per condition from 2 experiments. **g** Box plots showing the number of Polycomb bodies detectable for RING1B, PCGF2 and PCGF6 with (green) and without (blue) 96 hr dTAG-13 treatment. $n > 50$ cells per condition from 2 experiments. Indicated $p$-values were calculated using a two-tailed Student's $t$ test. Source data are provided as a Source Data file. **h** Box plots comparing the number of nuclear foci counted for PCGF2 with (green) and without (blue) 96 h dTAG-13 treatment. Foci were divided into quartiles based on focus volumes in untreated cells. $n > 50$ cells per condition from 2 experiments. Indicated $p$-values were calculated using a two-tailed Student's $t$ test. Source data are provided as a Source Data file. In **c–e** and **g–h**, boxes represent the interquartile range (IQR), the middle line corresponds to the median, and whiskers extend to the largest and smallest values no more than 1.5 x IQR from the box. Values outside of this range are not plotted, but are included in all analyses.

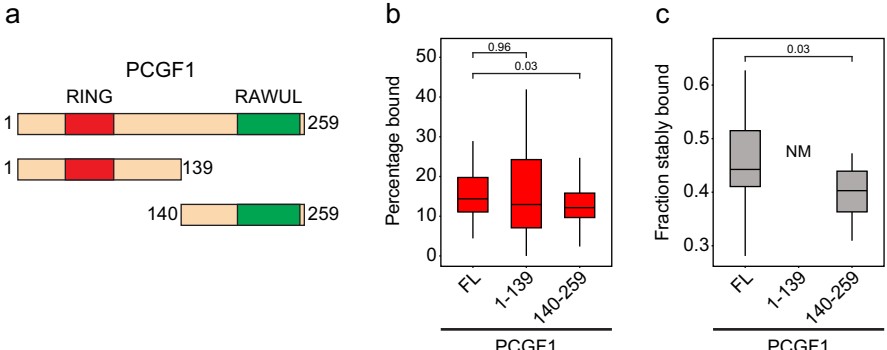

**Fig. 9 The C-terminal RAWUL domain of PCGF1 is required for stable chromatin binding. a** A schematic illustrating the PCGF1 constructs used for exogenous expression. The RING (aa 47–86, red) and RAWUL domains (aa 190–253, green) are indicated. **b** A box plot indicating the percentage of each PCGF1 fusion molecule (FL full length) in the bound state. Box plots represent measurements from $n = 90$ movies from 3 experiments each. Indicated $p$-values were calculated using a two-tailed Student's $t$ test. Source data are provided as a Source Data file. **c**. A box plot indicating the fraction each PCGF1 fusion molecule exhibiting stable binding. Stable binding of PCGF1 1–129 was not measurable (NM). Box plots represent measurements from $n = 16$ movies from 2 experiments each. The indicated $p$-value was calculated using a two-tailed Student's $t$ test. Source data are provided as a Source Data file. In **b–c**, boxes represent the interquartile range (IQR), the middle line corresponds to the median, and whiskers extend to the largest and smallest values no more than 1.5 x IQR from the box. Values outside of this range are not plotted, but are included in all analyses.

Strikingly, however, we could not detect quantifiable stable binding of the N-terminal fragment in 2 Hz survival time imaging, while the C-terminal fragment almost completely recapitulated the behaviour of the full-length protein (Fig. 9c). Interestingly, the absence of stable binding by the N-terminal fragment did not significantly affect its bound fraction, suggesting an inability to interact with KDM2B/BCOR/BCORL1 could possibly lead to more frequent but less stable binding events. Together, these observations indicate that stable chromatin binding by PCGF1 requires the C-terminal RAWUL domain, possibly through the chromatin binding activities of KDM2B and BCOR/BCORL, but not its RING1B dimerization domain. Therefore, in contrast to canonical PRC1 that requires interactions with H3K27me3 to increase the frequency but not stability of binding to chromatin, we discover that the RAWUL domain of PCGF1 enables stable binding, but does not define the frequency of binding. Together these observations indicate that auxiliary PRC1 complex components contribute in distinct ways to the kinetics of chromatin binding and that these important behaviours can only be uncovered using single-molecule live cell approaches.

## Discussion

Here, using endogenous protein tagging and single-molecule imaging, we systematically characterise the behaviour of PRC1 in live cells. We discover that PRC1 complexes are highly dynamic and only a small fraction binds stably to chromatin (Figs. 1 and 2). By quantifying the number of RING1B molecules in single cells, the fraction that are chromatin bound, and estimating their occupancy throughout the genome, we discover that a surprisingly small number of PRC1 complexes are bound at target genes (Fig. 3). We show that chromatin binding by PRC1 does not rely on the intrinsic nucleosome binding activity of its catalytic core, suggesting a central role for auxiliary proteins in chromatin binding by PRC1 (Fig. 4). By testing how different PRC1 complexes create the observed dynamics of PRC1, we discover that canonical PCGF2-PRC1 is most abundant, constitutes the majority of chromatin-bound PRC1, and is enriched in Polycomb bodies (Figs. 5 and 6). Conversely, we discover that variant PRC1 complexes, made up of PCGF1, 3, or 6, are less numerous and have distinct chromatin binding properties that correspond to their target site recognition and roles in depositing H2AK119ub1

(Fig. 7). Building on these findings, we perturb proposed PRC1 targeting mechanisms, and uncover a role for H3K27me3 in increasing the frequency with which canonical PRC1 binds chromatin, but not its capacity to bind stably to chromatin (Fig. 8). In contrast, we reveal that stable chromatin binding by the variant PCGF1-PRC1 complex relies on the RAWUL domain of PCGF1, which is known to interact with DNA binding factors (Fig. 9). Together, these findings provide a detailed quantitative understanding of the dynamics of PRC1 complexes and their interactions with chromatin in live cells, and reveal that a surprisingly small number of PRC1 complexes and low target gene occupancy is sufficient to support H2AK119ub1 and gene repression by the Polycomb system.

Although Polycomb repressive complexes have been extensively studied, the mechanisms by which they affect transcriptional repression have remained poorly understood[4,8]. PRC1 catalyses H2AK119ub1, and this has been implicated in transcriptional repression[79,80,88,108,109]. However, PRC1 protein complexes also occupy target genes and have been proposed to repress gene expression through a number of more direct and catalysis-independent mechanisms. These include limiting chromatin accessibility, supporting chromatin compaction/interaction, or possibly through phase separation[20–26,47,94,95]. Based on these observations, it has been debated whether PRC1 primarily uses H2AK119ub1 or catalysis-independent functions for transcriptional repression. If PRC1 primarily represses transcription via catalysis-independent mechanisms, one might expect a requirement for high-level physical occupancy of PRC1 at target genes. However, from our quantitative measurements of PRC1 subunit abundance and chromatin binding in live cells, we estimate there are a surprisingly small number of PRC1 complexes bound to target sites at any given time. This is also true if one focuses on canonical PRC1 complexes that are responsible for effects on chromatin compaction and architecture[20,21,23,26,27,47]. Furthermore, when we examine Polycomb bodies, which have been proposed to correspond to phase separated entities[110], the average number and concentration (10 molecules, 130 nM) of PRC1 complexes inside them is below that which is thought to be required to support phase separation[94,95,111]. Therefore, the sparseness of PRC1 occupancy at Polycomb target sites and in Polycomb bodies, suggests that structural effects and phase separation are likely not the primary determinants of PRC1-mediated gene repression in ESCs. These inferences are consistent with our previous findings that deletion of PCGF2/4, which disrupts canonical PRC1 complexes, has a limited effect on gene repression by PRC1 in this cell type[49] and that Polycomb complexes do not appear to limit chromatin accessibility[112]. Furthermore, it is consistent with the fact that CBX7, which in ESCs is the predominant CBX protein in canonical PRC1 complexes, does not exhibit phase separating behaviour in vitro[23,94]. Interestingly, it has also been reported in *Drosophila* that there are very low cellular concentrations of PRC1 complex components in the developing embryo[53,54,113]. Unless the stably bound fraction of molecules in these systems is much higher, it would also be difficult to rationalise how physical or structural effects via the occupancy of these complexes on chromatin could be the primary effector of PRC1-mediated gene repression.

In contrast to canonical PRC1, removal of variant PRC1 complexes in ESCs causes derepression of thousands of Polycomb target genes[49,91]. A central distinction between canonical and variant PRC1 complexes is their catalytic activity. Canonical PRC1 complexes have limited capacity to catalyse H2AK119ub1 in vitro and contribute virtually nothing to H2AK119ub1 in vivo[38,49,50]. In contrast, variant PRC1 complexes are highly active in vitro and account for almost all H2AK119ub1 in vivo. Catalytic mutation of PRC1 causes Polycomb target gene derepression, despite variant PRC1 complex occupancy at target sites being largely unaffected as measured by ChIP-seq[79,108]. This suggests that H2AK119ub1, as opposed to variant PRC1 complex occupancy, may play a central and potentially direct role in gene repression. In agreement with this possibility, we estimate at RING1B sites with the highest density of H2AK119ub1 that between 60-100% of nucleosomes have at least one H2AK119ub1. We envisage that this level of histone modification, even if dynamically turned over, would be compatible with persistent transcriptional repression. Therefore, we propose that H2AK119ub1, as opposed to PRC1 occupancy, is primarily responsible for PRC1-dependent gene repression in ESCs. In future work, it will be important to measure the number and kinetic parameters of canonical and variant PRC1 complexes during differentiation and in more committed cell types. This will allow us to understand whether the general principles we observe in ESCs are recapitulated, or if canonical PRC1 complexes play a more direct and active role in gene repression as has been reported previously[24,26,109,114,115].

Our single particle tracking experiments revealed that PRC1 complexes are for the most part highly dynamic. In the case of variant PRC1 complexes, this behaviour may be important for catalysis and the differences in the dynamics of individual variant PRC1 complexes may be required to achieve the appropriate distribution of H2AK119ub1 in the genome. For example, we previously demonstrated that H2AK119ub1 is present across the genome at low levels and that this is primarily dependent on PCGF3/5-PRC1 complexes[49,85]. PCGF3/5 proteins are not significantly enriched at Polycomb target genes[91] and SPT revealed that the PCGF3-PRC1 complex had the smallest bound fraction and shortest binding times of all PRC1 complexes. Therefore, we propose that PCGF3/5-PRC1 complexes may utilise a highly dynamic hit-and-run mechanism to deposit low levels of H2AK119ub1 indiscriminately throughout the genome. In contrast, PCGF1- and PCGF6-PRC1 complexes show more specific enrichment at gene promoters as measured by ChIP-seq[31,49,91]. This appears to be reflected in our SPT measurements where they display a higher bound fraction or more stable binding to chromatin than PCGF3-PRC1. This likely corresponds to the fact that PCGF1- and PCGF6-PRC1 complexes incorporate auxiliary subunits that have DNA or chromatin binding activities, which cause generic targeting to gene promoters. In agreement with this, removal of the C-terminal RAWUL domain of PCGF1, which interacts with chromatin binding subunits BCOR and KDM2B[30,33,40,48,116], disrupted stable chromatin binding by PCGF1. Therefore, in contrast to PCGF3/5-PRC1 complexes, we propose that PCGF1- and PCGF6-PRC1 complexes more frequently occupy gene promoters via their site-specific chromatin binding activities, which provides an opportunity to enrich H2AK119ub1 at these sites. Interestingly, however, most sites occupied by PCGF1- and PCGF6-PRC1 complexes do not display elevated H2AK119ub1 when analysed by ChIP-seq[49,91]. We have previously proposed that transcription may block deposition of H2AK119ub1[4,117], meaning that despite PCGF1- and PCGF6-PRC1 complexes binding to and sampling the majority of gene promoters, H2AK119ub1 would only accumulate at sites lacking transcription in order to maintain, as opposed to initiate, a transcriptionally repressed state.

In summary, our quantitative single molecule measurements of PRC1 dynamics and chromatin binding in live cells, coupled with estimation of genome-wide target site occupancy, provide important insight into how PRC1 interacts with chromatin. Furthermore, it provides additional evidence to suggest that H2AK119ub1, as opposed to PRC1 occupancy and non-catalytic functions, is the primary determinant of PRC1-mediated repression in ESCs.

## Methods

**Cell culture**. All mouse ESC lines used in this study were grown at 37 °C and 5% CO2 on gelatinised plates. Dulbecco's Modified Eagle Medium (Gibco) was supplemented with 15% foetal bovine serum (Labtech), 2 mM L-glutamine (Life Technologies), 1x penicillin-streptomycin solution (Life Technologies), 1x non-essential amino acids (Life Technologies), 0.5 mM beta-mercaptoethanol (Life Technologies), and 10 ng/mL leukaemia inhibitory factor. Cells were passaged using trypsin-EDTA (0.25%, Gibco) with 2% chicken serum. Cells were regularly tested for the presence of mycoplasma.

To induce conditional knockout, PRC1CKO cells were treated with 800 nM 4-hydroxytamoxifen (OHT) (Sigma) for 72 h. To induce and maintain degradation and depletion of SUZ12, dTAG-SUZ12 were treated with 100 nM dTAG-13[101] for 96 h.

**Alkaline phosphatase staining**. In order to assess pluripotency by AP staining, the PRC1CKO line, and the same line expressing either RING1B-HaloTag or RING1BNBM-HaloTag were plated and grown in the presence or absence of OHT for 72 h. 10,000 cells per condition were plated out on a 10 cm plate after counting using a Countess II cell counter (Invitrogen). Cells were grown for 4 days and then fixed with 2% paraformaldehyde. Fixed cells were stained using AP staining solution (100 mM Tris-HCl pH 9.0, 100 mM NaCl, 5 mM MgCl2, 0.4 mg/mL napthol phosphate N-5000, 1 mg/mL Fast Violet B salt) for 10 min, rinsed with PBS and distilled water, and then air-dried. Three biological replicates were carried out and at least 100 colonies were counted and classified for each condition in each replicate.

**Stable cell line generation by random integration**. To generate cell lines expressing H2B-HaloTag the corresponding cDNA was cloned into an expression plasmid driven by the CAG promoter that also contained an internal ribosome entry site that allowed co-expression of the G418 resistance gene (primers are listed in Supplementary Table 1). For HaloTag-3xNLS expression, the pHTN CMV-neo vector (Promega), modified with 3x SV-40 NLS at the C-terminus (gift, James Rhodes), was used. In order to randomly integrate this expression cassette, ESCs in one well of a 6-well plate were transfected with 10 μg of expression vector using Lipofectamine 3000 (ThermoFisher) according to the manufacturer's instructions. The following morning, transfected cells were passaged onto new plates at a range of cell densities. 5 h later, 400 μg/ml of G418 was applied. Cells were grown in the presence of G418 to select for stable integration events and approximately one week later resistant colonies were picked and expanded on 96-well plates. 96 well plates of putative positive clones were labelled with 500 nM Halo-TMR dye (Promega) for 15 min at 37 °C, washed in label free medium three times, and then incubated in medium without dye for 30 min at 37 °C. Individual clones with stable HaloTag fusion protein expression were identified by fluorescence microscopy.

**Endogenous tagging of genes via CRISPR/Cas9-mediated HDR**. In order to introduce tags into the endogenous copies of genes using CRISPR-mediated genome editing and HDR, sgRNAs were designed using the CRISPOR online tool[118] and cloned into pSptCas9(BB)-2A-Puro(PX459)-V2.0 (Addgene #62988)[119] (sgRNA sequences are listed in Supplementary Table 2). Targeting constructs for homology directed repair (HDR) were engineered to lack the guide sequence and were assembled from PCR products by Gibson assembly (Gibson Assembly Master Mix kit, New England Biolabs) (primers are listed in Supplementary Table 1).

To generate endogenous HaloTag or dTAG fusion cell lines, targeting constructs were generated to modify the endogenous locus such that it encoded the HaloTag or dTAG with a flexible polypeptide fused to either the N- or C-terminus of the gene of interest.

Targeting was carried out by transfecting 3 μg of each Cas9 guide and 10 μg of targeting construct into ESCs in one well of a 6-well plate using Lipofectamine 3000 (ThermoFisher) according to the manufacturer's instructions. The following morning, transfected cells were passaged onto new plates at a range of cell densities. 5 h later puromycin was applied to cells at 1 μg/ml to select for transfected cells.

After 48 h of selection puromycin was removed and ~1 week later individual clones were picked and expanded onto 96 well plates. Alternatively, cells were labelled with 500 nM Halo-TMR dye (Promega) as described above, and FACS sorted to enrich for cells expressing a HaloTag. FACS sorted cells were then plated at low density and approximately 1 week later individual clones were picked and expanded onto 96 well plates.

Appropriately edited clones were then identified by PCR screening from genomic DNA (primers are listed in Supplementary Table 1). A PCR screen with a primer pair flanking the targeting construct was used to amplify genomic DNA in clones that showed evidence of HDR to test whether they were heterozygous or homozygote. When compared to parental lines, lines that where HDR had occurred in both alleles produced a product that was longer by exactly the size of the inserted sequence and showed no evidence of a wild type PCR product. The integrity of homozygote clones was then further validated by sequencing the PCR products to ensure the expected HDR event had occurred, western blotting, and fluorescence microscopy for HaloTag cell lines.

**Stable expression of HaloTag fusion proteins through CRISPR/Cas9-mediated editing of the TIGRE locus**. To generate cell lines expressing RING1B, RING1B nucleosome binding mutant (NBM, R81A, K93A, K97A, K98A), RPCD, PCGF1, PGF1 1-139, and PCGF1 140-259 HaloTag fusion proteins, targeting constructs were engineered to insert an expression cassette into the TIGRE locus[120] (primers are listed in Supplementary Table 1). An sgRNA for the TIGRE locus was cloned into pSptCas9(BB)-2A-Puro(PX459)-V2.0 (Addgene #62988)[119] (the sgRNA is listed in Supplementary Table 2).

Targeting constructs contained the cDNA of interest cloned next to a CAG promoter and fused to the HaloTag sequence with a FLAG 2xStrep-tag II tag to enable immunoblotting. A triple SV40 NLS sequence was also included in the RPCD and all PCGF expression constructs to ensure they localised to the nucleus.

Targeting was carried out by transfecting 3 μg of each Cas9 guide and 10 μg of targeting construct into ESCs in one well of a 6-well plate using Lipofectamine 3000 (ThermoFisher) according to the manufacturer's instructions. The following morning, transfected cells were passaged onto new plates at a range of cell densities. After 5 h, selection was applied using 1 μg/ml puromycin. Selection was maintained for 48 h, and then surviving clones grown out.

HaloTag expressing cells were enriched by FACS as described above and colonies were transferred to 96-well plates. Clones that had acquired the expression cassette via HDR at the TIRGRE locus were identified by a PCR screen with a primer inside the expression cassette and outside of the targeting construct. Then a primer pair flanking the targeting construct was used amplify genomic DNA in clones that showed evidence of HDR to test whether they were heterozygotes or homozygotes as described above. To limit the level of exogenous expression heterozygous clones were selected and further validated by sequencing the PCR products to ensure the expect HDR event had occurred, western blotting, and fluorescence microscopy.

**Nuclear and histone extracts**. Cell pellets were resuspended in 10 volumes of buffer A (10 mM Hepes pH 7.9, 1.5 mM MgCl2, 10 mM KCl, 0.5 mM DTT, 0.5 mM PMSF, and cOmplete protease inhibitor (PIC, Roche)) and incubated for 10 min on ice. After centrifugation at 1,500 g for 5 min, the cell pellet was resuspended in 3 volumes of buffer A supplemented with 0.1% NP-40. Following inversion, nuclei were pelleted again at 1,500 g for 5 min. Pelleted nuclei were resuspended in 1 volume of buffer C (250 mM NaCl, 5 mM Hepes pH 7.9, 26% glycerol, 1.5 mM MgCl2, 0.2 mM EDTA, 0.5 mM DTT and 1x PIC). The volume of the nuclear suspension was measured and NaCl concentration increased to 400 mM by dropwise addition. Nuclei were incubated at 4 °C for 1 h to extract nuclear proteins, which were recovered as the supernatant after centrifugation at 18,000 g for 20 min. Protein concentration was determined by Bradford assay (BioRad).

Histone extracts were prepared by washing pelleted cells in RSB (10 mM Tris HCl pH 7.4, 10 mM NaCl, 3 mM MgCl2 and 20 mM NEM), followed by centrifugation at 240 g for 5 min and resuspension in RSB buffer supplemented with 0.5% NP-40. Following incubation on ice for 10 min, cells were centrifuged at 500 g for 5 min. The nuclear pellet was resuspended in 5 mM MgCl2, an equal volume of 0.8 M HCl was added, and incubated on ice for 20 min to extract histones. The supernatant was taken after centrifugation for 20 min at 18,000 g, and histones precipitated by addition of TCA up to 25% by volume and incubation on ice for 30 min. Histones were pelleted by centrifugation at 18,000 g for 15 min, and the pellet washed with cold acetone twice. The histone pellet was resuspended by vortexing in 1x SDS loading buffer (2% SDS, 100 mM Tris pH 6.8, 100 mM DTT, 10% glycerol and 0.1% bromophenol blue) and boiling at 95 °C for 5 min. For histone extractions where H2AK119ub1 was to be analysed, all buffers were supplemented with 20 mM NEM.

**Recombinant protein expression and purification**. Mouse RING1B, with or without a HaloTag, was cloned into a prokaryotic expression vector (pNIC28) that contains a 6-His tag for the purposes of purification (primers are listed in Supplementary Table 1). RING1B or RING1B-HaloTag expression vectors were transformed into the *E. coli* BL21 (DE3) pLysS strain. Following induction of protein expression, cultures were supplemented with 250 μM ZnCl2. Lysis was performed by sonication in buffer containing 20 mM Tris (pH 8.0), 500 mM NaCl, 0.1% NP-40 and cOmplete Protease Inhibitor Cocktail (Roche). RING1B or RING1B-HaloTag was purified from lysates using Ni²⁺-charged IMAC Sepharose 6 Fast Flow resin (GE Healthcare). Washes were performed with buffer containing 50 mM NaH2PO4 (pH 8.0), 300 mM NaCl and 50 mM imidazole and protein was eluted in the same buffer containing 250 mM imidazole. Elution fractions containing protein were pooled.

**Size exclusion chromatography**. Purified recombinant RING1B or RING1B-HaloTag and nuclear extracts were first dialysed into BC200 buffer (50 mM HEPES (pH 7.9), 200 mM KCl, 10% Glycerol, 1 mM DTT). Samples were separated on a Superose 6 Increase 10/300 GL column (GE Healthcare, precalibrated with dextran blue, Mix 1 (Ferritin, 440 kDa; Conalbumin, 75 kDa) and Mix 2 (Thyroglobulin, 669 kDa; Aldolase, 158 kDa; Ovalbumin, 43 kDa)) in BC200 buffer and collected in 250 μl fractions. 1% of the fractions for purified proteins were directly analysed by western blot. Nuclear extract fractions were first trichloroacetic acid precipitated and then 20% of the fraction was analysed by western blot.

**Immunoprecipitation**. Immunoprecipitations were performed using 500 μg of nuclear extract. Extracts were diluted to 550 μL with BC150 (150 mM KCl, 10% glycerol, 50 mM HEPES (pH 7.9), 0.5 mM EDTA, 0.5 mM DTT, 1x PIC) and incubated with antibody against the protein of interest overnight at 4 °C (antibodies are listed in Supplementary Table 3). Protein A beads (Repligen) were used to capture antibody-bound protein at 4 °C for 1 h. Beads were pelleted at 1000xg, and washed 3 times with BC300 (300 mM KCl, 10% glycerol, 50 mM HEPES (pH 7.9), 0.5 mM EDTA, 0.5 mM DTT). For western blotting, beads were resuspended in 2x SDS loading buffer and boiled at 95 °C for 5 min, and the supernatant taken as the immunoprecipitate. 1x SDS loading buffer was added to input samples which were also incubated at 95 °C for 5 mins, prior to SDS-PAGE and western blot analysis.

**Immunoblotting**. Protein extracts were mixed with SDS loading buffer and boiled at 95 °C for 5 min. Proteins were resolved by SDS-polyacrylamide gel electrophoresis (SDS-PAGE). Gels (0.1% SDS, 0.1% ammonium persulphate (Sigma), 0.1% TEMED (Sigma), 400 mM Tris HCl pH 8.8) were cast using the Mini-Protean Tetra Cell system (BioRad). Depending on the size of the protein of interest, resolving gels between 6% and 15% acrylamide/bis-acrylamide (BioRad) were used. The stacking gel contained 5% acrylamide/bis-acrylamide and 125 mM Tris HCl pH 6.8. Gels were run at 200 V in 1x SDS-PAGE running buffer (25 mM Tris, 192 mM glycine and 0.1% SDS). Proteins were transferred to nitrocellulose membrane by semi-dry transfer using the Trans-Blot Turbo Transfer System (BioRad, 1.5 A for 7 min). Membranes were blocked in 5% milk in 1x PBS with 0.1% Tween 20 (PBST, Fisher) for 30 min at room temperate. Primary antibody incubations were carried out overnight at 4 °C in the same buffer (antibodies are listed in Supplementary Table 3). Membranes were washed for 3×5 min with PBST, and then incubated with secondary antibody for 1 h in 5% milk in PBST. After 3×5 min PBST washes and one PBS rinse, membranes were imaged using an Odyssey Fc system (LI-COR).

**Fluorescence-based protein quantification**. We quantified the number of proteins in cells as described previously[81]. Briefly, cells were trypsinised, pelleted, and counted with a Countess II cell counter. $5 \times 10^6$ cells were labelled with 500 nM HaloTMR for 15 min at 37 °C which causes near-quantitative labelling of HaloTag proteins in live cells[121]. Cells were pelleted, counted again, and $3 \times 10^6$ labelled cells were reserved as a pellet. HaloTag Standard Protein (Promega) was labelled with 10 fold molar excess (5 μM) Halo-TMR dye at 4 °C for 15 min. Pelleted cells were lysed by boiling in 1× SDS loading buffer at 95 °C for 10 min and volumes equivalent to $1 \times 10^5$ and $2.5 \times 10^5$ cells were resolved by SDS-PAGE alongside known quantities of HaloTag Standard Protein. Gels were imaged using a Typhoon FLA 7000 (Fujifilm) using the 532 nm laser. Band intensities were quantified using Image Studio software (LI-COR). Calculated numbers of molecules and estimated nuclear concentrations are given in Supplementary Table 4.

**Single particle tracking**. Cells for single particle tracking were plated onto gelatinised and glycine-coated 35 mm petri dishes containing a 14 mm No. 1.5 coverglass (MatTek, #P35G-1.5-14-C) at least 5 h before imaging. Cells were labelled using 100 nM PA-Halo-JF$_{549}$ (gift, Luke D. Lavis and Jonathan B. Grimm)[67] for 15 min at 37 °C. They were then washed 3 times with Fluorobrite DMEM (Thermo Fisher Scientific) supplemented as described for general ESC culture above. Cells were incubated for a further 30 min in supplemented Fluorobrite DMEM at 37 °C and washed once more before imaging.

Single molecule tracking experiments were carried out using a custom TIRF/HILO microscope as described in[122]. The angle of the excitation beam is controlled by translation of the position of the focus in the objective (Olympus 100x NA1.4), and was positioned in each experiment to maximise the signal-to-noise ratio. Sample temperature was maintained at 37 °C using an objective collar heater and heated stage, while the pH of sample medium was maintained by addition of HEPES pH 7.4 to a concentration of 30 mM. For rapid tracking of diffusing and bound molecules, most movies were acquired with continuous 561 nm and 405 nm excitation at 22 mW and 5 mW intensity at the fibre output, respectively, with a 15 ms exposure time. 405 nm excitation intensity was modulated to ensure a low density of photoactivated fluorophores based on protein expression. Movies were acquired for 4000 frames and contained multiple nuclei. Experiments were performed in at least three biological replicates of at least 10 movies each, with each movie containing multiple cells.

To measure binding times of single molecules, movies were acquired with a frame rate of 2 Hz and exposure time of 0.5 s for 600 frames or at 0.03 Hz with an exposure time of 1 s for 30 frames. Acquisition was started after activating fluorophores using the 405 nm laser, avoiding densities of photoactivated fluorophores that would be too high to reliably track, and carried out with 561 nm excitation at 0.1 mW. Experiments were performed in at least two biological replicates of eight movies each, with each movie containing multiple cells.

**Confocal imaging**. Polycomb body imaging, FRAP experiments, and relative fluorescence intensity measurements were carried out with a Spinning Disk Confocal microscope. Cells were plated on gelatinised 35 mm petri dishes containing a 14 mm No. 1.5 coverglass (MatTek, #P35G-1.5-14-C) at least 5 h before imaging.

Prior to imaging, cells were labelled with 500 nm Halo-JF$_{549}$ (gift, Luke D. Lavis and Jonathan B. Grimm)[66] for 15 min at 37 °C, followed by 3 washes, changing medium to Fluorobrite DMEM (Thermo Fisher Scientific) supplemented as described for general ESC culture above. Cells were then incubated for a further 30 min at 37 °C in supplemented Fluorobrite DMEM to remove excess dye. For Polycomb body imaging, 10 μg/ml Hoechst 33258 (Thermo Fisher Scientific) was also added to the medium during this incubation. Cells were washed once more with Fluorobrite DMEM before imaging. Confocal microscopy was performed on an IX81 Olympus microscope connected to a spinning disk confocal system (UltraView VoX PerkinElmer) using an EMCCD camera (ImagEM, Hamamatsu Photonics) in a 37 °C heated, humidified, CO2-controlled chamber.

**Polycomb body imaging**. Z-stacks for imaging Polycomb bodies were acquired on the above described spinning disk confocal system using an Olympus PlanApo 100x/1.4 N.A. oil-immersion objective heated to 37 °C, and Volocity software (PerkinElmer). Z-stacks for Polycomb body quantification were acquired by imaging Halo-JF$_{549}$ with a 568 nm laser at 1.25 s exposure and 15% laser power, while Hoechst was imaged with a 405 nm laser at 250 ms exposure and 20% laser power. Z-stacks were acquired at 150 nm intervals. Experiments were performed in at least two biological replicates of at least 15 cells each.

**FRAP imaging**. FRAP experiments were performed on the above described spinning disk confocal system using an Olympus PlanApo 60x/1.4 N.A. oil-immersion objective heated to 37 °C, and Volocity software (PerkinElmer). Spot FRAP was performed by acquiring 10 prebleach frames at 1 s intervals, then bleaching a circle of either 2.5 μm (general FRAP) or 1.4 μm (Polycomb body FRAP) in diameter using the 568 nm laser at 100% power. For Polycomb body FRAP, spots were bleached in either the nucleoplasm or a Polycomb body. Post-bleaching images were acquired at 0.5 s or 1 s intervals for 10 min. Images were acquired using the 568 nm laser at 400 ms exposure and 15% laser power. Experiments were performed in two biological replicates of at least 10 cells each.

**Relative fluorescence intensity measurements**. Images to quantify relative fluorescence intensities within nuclei were acquired using the above described spinning disk confocal system with an Olympus PlanApo 60x/1.4 N.A. oil-immersion objective heated to 37 °C, and Volocity software (PerkinElmer). Single Z-slices of nuclei were acquired using the 568 nm laser at 400 ms exposure and 15% laser power. Nuclei were segmented manually, and mean fluorescence calculated and normalised to that of RING1B and the number of RING1B molecules determined as above. Calculated numbers of molecules are given in Supplementary Table 4.

**Immunofluorescence quantification of RING1B signal**. Wild type and RING1B-HaloTag cells were passaged together in a 1:1 ratio. Cells were trypsinised and labelled with Halo-TMR in suspension as described for fluorescence-based protein quantification above. Labelled cells were fixed with 3.3% formaldehyde for 10 min at room temperature. Permeabilisation (0.5% Triton X-100 in PBS), blocking (3% BSA in PBS for 30 min), primary (2.5 h, anti-RING1B, Cell Signalling Technology) and secondary antibody incubations (1 h, Supplementary Table 3) and DAPI incubation (0.1 μg/ml, 10 min) were all performed in suspension. Following all treatments, cells were resuspended in VECTASHIELD mounting medium (Vector Laboratories), and 10 μl of cells were flattened on a slide by applying pressure to a coverslip. Single Z-slices of nuclei were acquired using the above described spinning disk confocal system with an Olympus PlanApo 60x/1.4 N.A. oil-immersion objective and Volocity software (PerkinElmer). Signal from Halo-TMR in the 561 nm channel was used to differentiate wild type and RING1B-HaloTag cells. Signal from secondary antibody was acquired using the 488 nm laser at 10% laser power with 500 ms exposures. Nuclei were segmented manually and mean 488 nm channel fluorescence calculated for each cell line.

**Tracking and localisation of SPT data**. Analysis of SPT data was performed in MATLAB (MathWorks) using Stormtracker software[75]. Fluorescent molecules were detected in each frame based on an intensity threshold and localisations of molecules determined to 25 nm precision by fitting an elliptical Gaussian point spread function. Localisations in consecutive frames of movies within a radius of 768 nm were linked to form tracks. Localisations were permitted to be lost for single frames within tracks as a result of fluorophore blinking or loss of focus through a memory parameter. Mean numbers of tracks for each protein in different imaging modes are given in Supplementary Table 5.

**Calculation of apparent diffusion coefficients**. Tracks with at least 5 localisations were used to determine apparent diffusion coefficients ($D^*$) based on the mean-squared displacement (MSD) of the total track. $D^*$ was calculated by

$$D^* = \frac{\mathrm{MSD}}{(4dt)} - \frac{\sigma^2}{dt} \qquad (1)$$

where $dt$ is the time between frames, with correction for localisation uncertainty σ.

**Spot-on analysis of tracks**. Analysis of tracks to determine the fraction of molecules in bound and diffusing states was performed using the online Spot-On tool[69]. Briefly, Spot-On determines the fractions of and diffusion coefficients of tracked molecules in different states using a two- or three-state kinetic model. Spot-On models the distributions of measured displacements from tracks, accounting for localisation error and correcting for bias due to movement of diffusing molecules out of focus. The probability of molecules diffusing out of focus is calculated and losses of diffusing molecules used to infer the fractions and diffusion coefficients for the model states. Jump length distributions were generated using a bin width of 0.01 μm with 8 time points and 4 jumps permitting a maximum jump length of 5.05 μm. Jump length distributions were fitted with a three-state kinetic model with a localisation error of 40 nm, using Z correction with $dZ = 0.7$ and cumulative density function fitting with three iterations. Tracks from individual movies, each containing multiple cells, were fitted separately. All statistical analyses were performed on individual movie fits. Bound fractions and estimated numbers of bound proteins for PRC1 subunits are given in Supplementary Table 4.

**Single molecule binding time analysis**. Tracking movies for measuring single molecule binding times were localised and tracked as described above, using a tracking radius of 192 nm for 2 Hz imaging or 672 nm for 0.03 Hz imaging. These limits were determined based on the measured displacements of stably bound H2B-HaloTag molecules due to chromatin diffusion. Track lengths of stationary molecules were used to determine apparent dwell times of chromatin bound molecules. Mean survival times were determined by fitting the observed distribution of apparent dwell times with a double exponential function:

$$y = \frac{Ae^{-\frac{t}{\tau_1}}}{e^{-\frac{t_1}{\tau_1}}} + \frac{(1-A)e^{-\frac{t}{\tau_2}}}{e^{-\frac{t_1}{\tau_2}}} \qquad (2)$$

where $y$ is the fraction of molecules remaining at time $t$, time $t_1$ represents the first time point, $A$ is the fraction of molecules with mean dwell time $\tau_1$, and $1–A$ is the fraction of molecules with mean dwell time $\tau_2$. In order to estimate binding times from these measurements, mean survival times were corrected for events that result in loss of tracks from reasons other than unbinding, such as photobleaching of dye molecules or diffusion of chromatin out of focus[72–75]. This was done by measuring the survival time for H2B-HaloTag, which is stably bound for hours and will therefore only rarely be observed to unbind, with all other tracks being limited by photobleaching or chromatin diffusion. The survival time of H2B-HaloTag, $t_{bleach}$ (typically approximately 50 s), was measured for each experiment to account for changes in imaging conditions. Binding times were calculated from fitted dwell time constants as follows:

$$t_{bound} = \frac{t_{dwell} \times t_{bleach}}{t_{bleach} - t_{dwell}} \qquad (3)$$

When $t_{dwell}$ exceeded $t_{bleach}$, or was close enough to $t_{bleach}$ such that $t_{bound}$ was more than double $t_{bleach}$, binding times were not estimated and are indicated as likely exceeding 100 s. Estimated stable binding times and fractions of observed binding events which were stable for PRC1 subunits are given in Supplementary Table 4.

**Calculation of target search time**. Target search times for PCGF3 were calculated as described previously[62,75] using the equation below:

$$\frac{N_{bound}}{N_{bound} + N_{free}} = \frac{t_{bound}}{t_{bound} + t_{search}} \qquad (4)$$

where $N_{bound}$ is the number of molecules in the bound state of interest and $t_{bound}$ is the mean time spent in this bound state. $N_{bound}$ was calculated for each bound state by multiplying the total number of molecules by the fraction of observed binding events belonging to that state and the fraction of all molecules that were bound. For PCGF3, because short binding events (1.5 s) predominate, and stable binding events are rare, significantly longer, and therefore highly infrequent (<4% of PCGF3 molecules, $t_{bound} = 39$ s, $t_{search} > 1000$ s), only the short binding time was taken into account when calculating $t_{search}$ in the context of catalysis.

**FRAP analysis**. To measure fluorescence recovery, intensity measurements were made with Fiji and ImageJ[123]. Fluorescence intensity was measured in the bleached region and a corresponding unbleached region within the same cell. From each, the intensity of a background region outside of cells was subtracted. The relative intensity of bleached and unbleached regions was calculated by dividing background-corrected unbleached intensity by background-corrected bleached intensity. Prebleach measurements were used to normalise all values so that the mean of the relative prebleach intensity was 1. The mean, normalised relative intensity of all repeats was plotted with error bars indicating standard error of the mean.

**Polycomb body analysis**. To segment Polycomb bodies in individual live cells for analysis, nuclei were first manually segmented in 3D based on Hoechst fluorescence in the 405 nm channel using TANGO in ImageJ[124]. The 561 nm channels of the same Z-stacks were deconvolved using Olympus cellSens software (constrained iterative deconvolution, 5 cycles). Deconvolved 561 nm z-stacks were masked using

outputs from TANGO, and individual Polycomb bodies identified as follows. Briefly, segmented nuclei were background subtracted using a 4 px rolling ball and a mask of Polycomb bodies generated using Otsu thresholding. The 3D Objects Counter plugin in ImageJ was used to quantify the properties of the masked Polycomb bodies, and its outputs (object volume, number, mean and total intensity) were processed and analysed using a custom R script to compile all data and calculate any values derived from the 3D Objects Counter measurements (fluorescence enrichment in foci, fractions of total nuclear volume and fluorescence in foci). The mean background intensity of each image stack, measured from a region containing no cells, was subtracted from all measured fluorescence intensities from the same stack. Apparent foci with volumes larger than 1.5 μm³ were excluded from measurements, as these likely resulted from segmentation identifying regions of the nucleus with fluctuations in fluorescence intensity outside of Polycomb bodies. Similarly, foci with volumes smaller than 0.029 μm³ were also excluded as they could not be accurately segmented. Quartiles of foci by volume or mean fluorescence intensity were assigned based on the distribution of foci in untreated cells, and quartile boundaries were used to classify foci in treated cells. Enrichment of signal within Polycomb bodies was calculated for each cell by comparison of mean fluorescence densities in all foci to mean fluorescence density for the rest of the nucleus. Estimates of numbers and concentrations of molecules in foci were made by determining the fluorescence intensity equivalent to a single molecule based on total nuclear fluorescence signal and measured numbers of proteins per cell (see above), and then taking into account the total fluorescence signal and volume of each Polycomb body, as below:

$$\text{Molecules per body} = \frac{\text{Molecules per nucleus} \times \text{Fraction of total signal in bodies}}{\text{Number of bodies in nucleus}} \qquad (5)$$

$$\text{Concentration in body} = \frac{\text{Molecules per body} \times N_A}{\text{Body volume}} \qquad (6)$$

**Estimating the number of H2AK119ub1 molecules and PRC1 catalytic activity**. To estimate the number of H2AK119ub1 molecules and the rate of catalysis by PCGF3/5-PRC1 we first estimated the number of histone H2A molecules in nucleosomes in a 2n mouse genome. To achieve this we used previously published[125] genome-wide measurements of the mean linear distance between nucleosomes (186.1 bp) in mouse ESCs and the size of the 2n mouse genome (5,461,710,950 base pairs) to estimate the total number of nucleosomes as follows:

$$\text{Total 2n nucleosomes} = \frac{\text{2n genome size}}{\text{Nucleosome spacing}} \qquad (7)$$

Based on this calculation we estimate that there are $2.9 \times 10^7$ nucleosomes. In each nucleosome there are 2 histone H2A molecules, meaning there are $5.9 \times 10^7$ histone H2A molecules. In ESCs ~10% of H2A is monoubiquitylated on K119[49] so we estimate there are $5.9 \times 10^6$ molecules of H2AK119ub1. In mammalian cells with a similar fraction of H2AK119ub1, the decay half time of this modification was previously determined to be 90 min[98]. On this basis, we estimated a decay constant of $1.3 \times 10^{-4}$ s$^{-1}$ which would correspond to loss of 750 H2AK119ub1 modifications per second, and the same rate of replacement by PRC1, assuming the system exists in a steady state. Based on previous findings that PCGF3/5-PRC1 contribute approximately half of global H2AK119ub1[49], we calculated how frequently each PCGF3/5-PRC1 complex should deposit H2AK119ub1 to maintain this rate, assuming approximately equal levels of PCGF3 and PCGF5 (total 6,000 molecules):

$$\text{Frequency of deposition by PCGF3/5}$$
$$= 1 \div \left( \frac{\text{Deubiquitylation rate}}{2} \times \frac{1}{\text{Total PCGF3/5 molecules}} \right) \qquad (8)$$

On this basis we estimate that each PCGF3/5 complex should deposit a H2AK119ub1 modification every 16 s to maintain H2AK119ub1 levels.

**Analysis of ChIP-seq data**. Aligned and processed ChIP-seq data for RING1B, PCGF1, PCGF2, PCGF6, CBX7 and H2AK119ub1 in ESCs was obtained from previously published work[49,79] (GEO: GSE132752 and GSE119618, Supplementary Table 6). As described previously[79], RING1B peaks (18,643) correspond to regions of the genome that have a peak of RING1B that is lost following conditional removal of RING1B as quantified using calibrated ChIP-seq analysis. Paired-end reads were previously[79] aligned to the mouse genome (mm10) using Bowtie 2[126] with "-no-mixed" and "-no-discordant" options, and only uniquely aligned reads after removal of PCR duplicates with Sambamba[127] were used. To visualise ChIP-seq distributions, genome coverage tracks were generated using the pileup function from MACS2[128] and viewed using the UCSC genome browser[129]. Read counts at RING1B peaks were extracted using multiBamSummary from deeptools ("–outRawCounts")[130].

**Estimation of maximal protein occupancy**. To estimate the maximal number of molecules that could occupy RING1B peaks, we used total molecule number measurements from biochemical experiments, the fraction of bound molecules from SPT experiments, and read counts in RING1B peaks from ChIP-seq

experiments. To estimate the number of chromatin bound molecules, the bound faction from SPT experiments was multiplied by the total number of molecules. The number of bound molecules at each peak was estimated as follows:

$$\text{Bound molecules in peak} = \text{Reads in peak} \times \frac{\text{Total bound molecules}}{\text{Reads in all peaks}} \quad (9)$$

The density of molecules per kilobase was estimated by dividing the number of molecules in a given peak by the length of the peak. Read counts and calculated densities for all peaks are available in Supplementary Data 1. We would also like to emphasise that our calculations will inevitably result in an overestimation of the number of molecules bound at RING1B peaks due to the two central assumptions that were made in order to capture what we would consider to be the maximal possible occupancy. The first assumption was that binding events would be restricted to RING1B peaks. The limitation of this assumption is that PRC1 must also bind chromatin outside of ChIP-seq peaks as H2AK119ub1 occurs, albeit at reduced levels, throughout the genome[49,85]. The second assumption we made was that each RING1B peak is present as two copies because ESCs are diploid (2n). However, our ESC cultures are not synchronous, so some cells will have replicated their DNA but not yet divided, meaning that a subset of cells will have DNA content greater than 2n. Despite these assumptions, and the overestimations that will inevitably be inherent to them, we reasoned that our maximal estimates would be useful in contextualising possible PRC1 functions at target sites, particularly given the structural roles that PRC1 is proposed to have at these loci.

**Estimation of H2AK119ub1 levels**. In order to estimate the number of H2AK119ub1-modified nucleosomes at RING1B peaks we multiplied the estimated total number of H2AK119ub1 molecules (see above) by the fraction of the genome that can be uniquely mapped using our sequencing strategy $(0.88, 2.4 \times 10^9 \text{ bp})$[131] and then multiplied this by the fraction of H2AK119ub1 ChIP-seq reads that fall in RING1B peaks as follows:

$$\text{H2AK119ub1 in peaks} = \text{Total H2AK119ub1} \times \text{Mappable genome fraction}$$
$$\times \frac{\text{Reads in peaks}}{\text{Total reads}} \quad (10)$$

Based on this calculation we estimate that there are $3.3 \times 10^5$ H2AK119ub1 molecules in RING1B peaks. We then estimated the number of H2AK119ub1 modifications at each peak as follows:

$$\text{H2AK119ub1 at peak} = \text{Reads at peak} \times \frac{\text{Total H2AK119ub1} \times \text{Mappable genome fraction}}{\text{Total reads}} \quad (11)$$

The density of H2AK119ub1 modifications per kilobase at peaks was calculated by dividing the estimated number of H2AK119ub1 modifications by the peak length.

**Reporting summary**. Further information on research design is available in the Nature Research Reporting Summary linked to this article.

## Data availability

The ChIP-seq datasets analysed in this current study are available in the GEO repository, with the accession codes GSE132752 and GSE119618. Read counts at RING1B peaks from the ChIP-seq datasets used are available in Supplementary Data 1. Source data are provided with this paper.

## Code availability

The macros and scripts used for segmenting and analysing Polycomb bodies in this study have been deposited and are available at [https://github.com/MKHuseyin/Polycomb-Body-Analysis].

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

## Acknowledgements

Work in the Klose lab is supported by the Wellcome Trust (109102/Z/15/Z to M.K.H, 209400/Z/17/Z to R.J.K.), the European Research Council (681440), and the Lister Institute of Preventive Medicine. We thank James Rhodes for his valuable guidance, suggestions, and assistance in establishing live cell microscopy and analysis techniques. We thank Aleksander Szczurek, Mathew Stracy, and Stephan Uphoff for their contributions to the analysis of microscopy data, and Jessica Kelley for her assistance in setting up and carrying out SEC experiments. We are grateful to Luke Lavis and Jonathan Grimm for their generous gift of the JF$_{549}$- and PA-JF$_{549}$-HaloTag ligands. We gratefully acknowledge the Micron Advanced Bioimaging Unit (supported by Wellcome Strategic Awards 091911/B/10/Z and 107457/Z/15/Z) for their support and assistance with all practical microscopy in this work. We thank Anders Hansen, Nadezda Fursova, Neil Blackledge and Paula Dobrinić for critical reading of the manuscript.

## Author contributions

Conceptualization, M.K.H and R.J.K.; Methodology, investigation, formal analysis and resources, M.K.H.; Writing, M.K.H. and R.J.K.; Funding acquisition and supervision, R.J.K.

## Competing interests

The authors declare no competing interests.
