## [Peer Review File · Nature Communications]

REVIEWER COMMENTS

Reviewer #1 (Remarks to the Author):

PRC1 complexes function as an essential chromatin-based repressor of gene transcription. However, how PRC1 can target genes to be repressed and achieve gene repression remains unclear. Using endogenous protein tagging and single-molecule imaging, Huseyin and Klose systematically investigated the behavior of PRC1 in living mESCs. They found that PRC1 complexes are highly dynamic and only a small fraction of the complexes bind stably to chromatin. They claimed that a surprisingly small number of PRC1 complexes and low target gene occupancy is sufficient to support H2AK119ub1 and gene repression by the Polycomb system. While it was not so easy to follow the various kinds of presented data, overall the paper is interesting and informative to understand how PRC1 complexes can silence certain genome regions. For publication in Nature Communications, several critical points should be addressed:

Major comments:

1. Line 131, "HaloTag-H2B had a bound fraction of..." This bound fraction value seems to be too low and inconsistent with FRAP data of H2B (Ref. 71): the free fraction is ~5%. Furthermore, Ref. 69 (Fig. 4H) showed that the total bound fraction of H2B-Halo is ~75-80%. I wondered how accurately the particles of RING1B-HT and H2B were tracked. Some validation data should be provided.

2. How many particles of RING1B-HT (in a single cell) were analyzed? The number of particles analyzed and typical video data for each experiment would be very useful for a better understanding.

3. Line 171, "Polycomb bodies accounted for just 1.3% of the total nuclear volume..." Correct volume estimation of Polycomb bodies by fluorescence may be difficult because of the optical resolution limitation unless their body sizes are large enough.

4. Line 274, "We estimate that on average each Polycomb body has 9 PCGF2 and 10 RING1B molecules..." I wondered how the authors estimated the average numbers: 10 RING1B molecules for each Polycomb body.

5. Fig. 2I and Fig. S4B. While the SPT demonstrated that 20% of RING1B was bound to chromatin, the FRAP data showed that ~40% of RING1B bound to chromatin stably in Polycomb bodies. It would be nice to see the RING1B behaviors in Polycomb bodies by SPT and also the RING1B localization by STORM. I wondered whether Polycomb bodies might have more stably bound PRC1 than the authors estimated.

6. Fig. 8B-E. The rapid depletion experiments using dTAG system is very nice, but dTAG-13 treatment for 96 hrs seems to be too long because indirect effects might be negligible. In addition, their differences (with and without dTAG-13) on the bound fractions are not so convincing (Fig. 8C and D).

Minor comments:

1. Based on subunit composition, PRC1 complexes are grouped into canonical and variant forms. For general readers, it might be helpful to have some schemes for them.

2. Line 152, "38% of observed binding events were stable..." I wondered where the value came from?

3. Line 190, "To estimate the maximum number of molecules that could possibly bind to RING1B-occupied target sites identified in ChIP-seq (18,643 sites)..." Since ChIP-seq data normally come from many cells, it is possible that actual RING1B-occupied target sites in the single-cell might be much fewer.

4. Line 322. The sentence has no period.

5. Fig. S1A. What does Panel A show? I guessed displacement histograms. A proper explanation of Spot-On analysis is needed so that the readers do not have to go to Ref. 69. Figure legends for A and B are in reverse order?

6. Fig. S2A Legend. What was alkaline phosphatase (AP) staining for? A differentiation marker? A proper explanation is required for general readers.

7. Fig.8B. The H3 bands are too faint.

Reviewer #2 (Remarks to the Author):

Polycomb group proteins are important for the maintenance of cellular identity. However, how they are targeted to chromatin and how long they spend there is not fully understood. The authors investigate the dynamics of Polycomb Repressive Complex 1 (PRC1) using single particle tracking and report that only a small fraction is chromatin bound. They take on the interesting but challenging task of estimating the number of PRC1 molecules found at bound chromatin sites. There are numerous different 'flavours' of PRC1 and the authors deliver an impressive effort in dissecting the dynamics of a variety of these 'flavours'. The tagging of endogenous alleles with HaloTags/dTags is elegant and should be commended. The manuscript is very well written and while some of the findings are based on assumptions/estimations, they are nonetheless thought-provoking and will be of value to the Polycomb field.

Major points:

1. Given that some PCGF2 can incorporate into variant PRC1 complexes (Tavares et al. 2012; Gao et al. 2012), the authors could have extended their analysis in Figure 6C and D as well as S4 to include CBX7, which would allow them better focus on canonical PRC1.
2. If possible, the authors could repeat the analyses in Figure S4 for RYBP to see if there is a differential enrichment of canonical and variant PRC1 at Polycomb bodies. The fact that PCGF2 is more enriched at Polycomb bodies than RING1B (Figure 2I and J) suggests that there will be a difference between canonical and variant PRC1 enrichment. However, CBX7 (canonical PRC1) and RYBP (variant PRC1) enrichment at Polycomb bodies would be a more suitable comparison to address this question.
3. In Figure 8, dTAG-SUZ12 could have been introduced into the cell line expressing HaloTagged CBX7 to focus on the contribution of H3K27me3 to canonical PRC1 dynamics exclusively.

Minor points:

1. In Figure 1C and 5B, if available, the authors could include a z-slice from WT cells (lacking HaloTag fusions) treated with JF549 as negative controls to highlight the specificity of the staining in the HaloTag fusions.
2. The authors do western blotting to show that they have successfully HaloTagged their proteins of interest (Figure 1A and S3). For the benefit of those who may not be very familiar with HaloTags, they might mention the expected size of the tag is 33kDa in legend of Figure 1A so that readers can appreciate that the expected size shift takes place. For some of the Halo-tagged proteins, there are some unfortunately sized background bands (i.e. for PCGF2 and PCGF3). In the Methods section, the authors write that, "The integrity of homozygote clones was then further validated by sequencing the PCR products to ensure the expected HDR event had occurred". If available, perhaps some of this sequencing data could be included as supplemental data.
3. The authors estimate that, "there would be on average 0.1 RING1B molecules for every kilobase of RING1B-enriched chromatin." They also estimate that, "sites in the top decile of RING1B density would still have fewer than 0.3 molecules per kb" and furthermore, they report that these are overestimations. These estimates are surprisingly low. Could the authors please comment on whether there are any technical limitations or caveats to the protocols used which could result in

an underestimation of the amount of RING1B molecules on chromatin.

4. In Figure 4, 6, 7 and 8 – how exactly are 'bound' and 'stably bound' molecules defined? Perhaps the authors could make this distinction more obvious in the text.

5. In the text, the authors write that 3.6% of PCGF2 is found in Polycomb bodies.

I'm confused as to how this figure was calculated. Earlier in the manuscript they write "our measurements revealed that Polycomb bodies accounted for just 1.3% of the total nuclear volume and that RING1B fluorescence signal inside Polycomb bodies was only 1.3-fold more intense than the surrounding nucleus (Figure 2J). This means that only 1.7% of total PRC1 signal originates from Polycomb bodies". This suggested to me that 1.3×1.3 was used to reach the 1.7% value. S4A suggests that PCGF2 signal is ~ 2.3 more intense in Polycomb bodies and 2.3×1.3 would equal to 3% (rather than 3.6%). Apologies, if I've misinterpreted something here.

6. The authors estimate that PCGF3/5-PRC1 complexes deposit a H2AK119ub modification every 16 seconds in order to maintain H2AK119ub levels. A prior single particle tracking study reported a "target search time" for CBX7 (Tatavosian et al. 2018). With the data the authors have collected, would it be possible to calculate a target search time for PCGF3/5? If possible, this could potentially support their 16 second estimation.

Reviewer #3 (Remarks to the Author):

This paper examines Polycomb repressive complex 1 (PRC-1), a well-known chromatin-interacting transcriptional repressor. The authors used CRISPR/Cas9 to genetically engineer a stable cell line integrating a HaloTag into the Ring1b gene, a component of the core subunit of PRC1 canonical and variant complexes. Single-particle-tracking was performed on RING1B-HaloTag, revealing 3 diffusing populations of PRC1. Using H2B-Halo as a control, only a small amount of PRC-1 complex is determined to be chromatin bound. A mathematical approach in comparison with ChIP data is used to estimate PRC-1 occupancy, demonstrating low occupant density at PRC-1 target sites. The authors track a non-functional mutated RING1B, and quantification of bound molecules is similar to wild-type protein. The authors conclude that interactions between the catalytic core of PRC1 and the nucleosome does not contribute significantly to chromatin binding and instead propose auxiliary proteins are responsible.

The authors continue by CRISPR-editing HaloTags into known proteins in variant PRC-1 complexes. Fluorescence quantification reveals various levels of expression, with PCGF2-PRC1 being the most abundant. SPT of variant complexes revealed distinct dynamics of diffusion and binding behavior. A dTAG system was then used to interrupt the placement of H3K27me3 across the genome. H3K27Me3 reduction did not have an effect on binding stability, but did slightly reduce the bound fraction of RING1B and PCGF2. Finally, SPT examination of the binding behavior of N- and C-terminal fragments of PCGF-1 demonstrated that the C-terminal fragment is responsible for binding behavior.

Overall, this is a nice paper demonstrating the quantitative power of single particle tracking when combined with CRISPR genome editing and ChIP-Seq. An impressive number of genes were CRISPR'd and the resulting SPT data are of high quality and rigor. Since the authors quantify such a low target-site occupancy for PRC1, especially the PCGF3/5-PRC1 variants responsible for most H2AK119ub1 deposition and gene repression, they conclude that H2AK119ub1 must be deposited in a hit-and-run like fashion (they estimate that each of the ~ 6000 PCGF3/5-PRC1 complexes deposits one H2AK119ub1 mark every ~ 16 seconds). This suggests the H2AK119ub1 modification itself, rather than direct occupancy by PRC1, accounts for most of the PRC1-mediated gene repression in ESCs.

Major Points

1. How much of RING1B is in complex? The authors use RING1B as a proxy for PRC-1 behavior, and demonstrate that HaloTag does not disrupt normal complex formation. However, if a significant amount of RING1B exists outside of PRC-1 complex, then analysis of RING1B SPT behavior would not provide an accurate representation of PRC-1 behavior. The authors should clarify this issues and tone down the text to make it clear that the dynamic they measure is a

combination of subunit assembly kinetics and binding kinetics.

Expanding on the above – an equally valid interpretation is that the majority of RING1B is not in complex, but in complex, the bound fraction is much higher. That is, the majority of observed bound fraction is actually when RING1B is in complex. Other data within the paper also alludes to the fact that complex formation is more important than individual binding (RING1BNBM mutant binding is reduced by a small but significant amount, and mutant which does not interact with PRC1 auxiliary factors has same binding as negative control).

2. Fig. 2F,G (related to Line 155 30 s interval distribution): Could it be that these are spots where more than one RING1B molecule came in and out? Is the density of fluorescent molecules at any given time low enough to rule this out? To rule out the possibility that long binding events are due to repeat binding, it would be good to show the intensity through time. The intensity should remain constant and photobleach in one step at the end of the measurement.

Minor Points

2. Full, uncropped gels for examination? I have no reason to doubt the veracity of the authors gels, but it is good practice that all uncropped gels images are provided for examination.

3. Fig. 2H – Is this a maximum intensity projection?

4. Line 359-360 – Full loss of H3K27me3 is not what their data shows, but a large reduction does seem to be observed.

Reviewer #4 (Remarks to the Author):

In this paper, the authors have endogenously tagged in mice embryonic stem cells different proteins that form the PRC1 complex. Using a HaloTag, the expression level, localization and dynamics of these proteins have been characterized. Notably, the authors have performed single protein tracking to monitor the fraction of proteins that strongly associates to the chromatin versus the fraction that freely diffuses in the nucleus. Using different mutations or protein depletion strategies, they have attempted to perturb the association of the protein of interest with the chromatin in order to understand the mechanisms of binding of the different members of the complex. While the manuscript is clearly written and presents interesting and surprising insights on the highly dynamic nature of the PRC1 complex, I have a few comments that I hope the authors can address.

The authors have used HaloTag-H2B and HaloTag-NLS as control for their SPT experiments. However, with the H2B control, they observe 60% of bound molecules, while FRAP data in the literature observe 90% of stably bound H2B (for instance: Kimura JCB 2001). Conversely, for a freely diffusing HaloTag-NLS, 11% of bound protein is measured. Where do those discrepancies come from? I would suggest performing additional experiments with control cells bearing no HaloTag to check if the Halo dye can interact non-specifically with some proteins or whether the washing step leaves unbound dye in the cells. FRAP experiments comparing HaloTag vs GFP tagged H2B or NLS could also help clarify this problem.

The authors have used very low imaging frequencies to monitor the stable binding events of a single molecule to the chromatin. How can the authors verify that what they observe represents the persistent binding of the same molecule and is not the exchange of multiple freely diffusing species on a stable structure?

In most of the figures, the authors present their data as the mean and the SEM of three biological replicates. However, there is a large diversity in the single-particle traces measured and I wonder if there isn't a lot of interesting information that could be extracted from this variability.

In the experiment performed with the mutant of RINGB that cannot bind to nucleosomes, the endogenous RINGB is still present. Can the presence of the functional endogenous protein affect the dynamics of RINGBNBM? I suggest that the authors try to measure the behavior of the mutant allele in absence of the endogenous protein.

The tagging of the various members of the PRC1 complex have been performed at the N or C

terminus. Is there a specific reason for these different tagging strategies and can it explain some of the differences measured between the PCGF proteins?

On line 274, the author claim that the FRAP dynamics recovered for PCGF2 are similar to the ones for RINGB. From a visual comparison of the two graphs, I would say that their behaviors are not similar. I don't think they have the same mobile fraction or that the kinetics of the fast recovering species are close to each other. A quantitative assessment of the kinetic parameters would be needed. If the FRAP curves are indeed different, it becomes difficult to explain why the SPT results between RINGB and PCGF2 are so similar.

The authors claim that the Polycomb bodies are mostly composed of canonical PRC1 complexes. However, their images also show an enrichment of PCGF1,3,6 in these structures. I would suggest measuring the enrichment of these proteins in these foci in order to compare the abundance of each protein in the Polycomb bodies. Dual-labeling of the cells could also provide very interesting insights, but is probably outside of the scope of this paper.

In Figure 9B and C, the full length and N-terminal fragment of PCGF1 display the same bound fraction, but for the N-terminal fragment no stably bound complex can be observed. I would suggest plotting the 1-CDF curves for these traces in order to better document this phenomenon. Mechanistically, I'm not sure how this behavior can be explained because intuitively I would have expected that in absence of the stably bound fraction the overall bound percentage would decrease. Additional FRAP measurements could also strengthen this observation.

Side comments

Because many figures and arguments in the paper discuss the percentage bound, the fraction stably bound and the stable binding time a more comprehensive description of how those measurements represent biologically and how they are extracted from the SPT data would be a nice addition.

In Figure 8H, a legend should be added next to the graph

The legends of Figure S1 A and B are inverted.

Serge Pelet

We thank the reviewers for their very supportive comments and suggestions to improve our manuscript. We have now carried out a series of new experiments and analysis to address these constructive comments and have updated the text and figures accordingly (altered text is highlighted red in the revised manuscript). Furthermore, bar graphs have now been replaced with box plots as required for publication in Nature Communications. Below we have provided a point-by-point response (black text) to the reviewer's comments (blue text). We believe these new experiments, analysis, and revisions have substantially improved the manuscript, and therefore we thank the reviewers for their time and extremely helpful input.

Reviewer #1: PRC1 complexes function as an essential chromatin-based repressor of gene transcription. However, how PRC1 can target genes to be repressed and achieve gene repression remains unclear. Using endogenous protein tagging and single-molecule imaging, Huseyin and Klose systematically investigated the behavior of PRC1 in living mESCs. They found that PRC1 complexes are highly dynamic and only a small fraction of the complexes bind stably to chromatin. They claimed that a surprisingly small number of PRC1 complexes and low target gene occupancy is sufficient to support H2AK119ub1 and gene repression by the Polycomb system. While it was not so easy to follow the various kinds of presented data, overall the paper is interesting and informative to understand how PRC1 complexes can silence certain genome regions. For publication in Nature Communications, several critical points should be addressed:

We appreciate the reviewer's supportive comments and for pointing out that our study was informative and interesting.

Major comments:

1. Line 131, "HaloTag-H2B had a bound fraction of..." This bound fraction value seems to be too low and inconsistent with FRAP data of H2B (Ref. 71): the free fraction is ~5%. Furthermore, Ref. 69 (Fig. 4H) showed that the total bound fraction of H2B-Halo is ~75-80%. I wondered how accurately the particles of RING1B-HT and H2B were tracked. Some validation data should be provided.

We thank the reviewer for highlighting this important point about H2B-HaloTag. Endogenous histones are almost exclusively expressed and incorporated into chromatin during S-phase¹⁻⁴. In our experiments the exogenous H2B-HaloTag expression cassette is randomly integrated into the genome and transcribed from a non-histone promoter that will lead to production of H2B-HaloTag throughout the cell cycle. H2B-HaloTag expressed outside of S-phase will not be immediately integrated into chromatin and its expression level can influence the overall unbound (non-incorporated) fraction. Despite the fact that we used similar expression approaches as in Ref 71 and Ref 69, there will inevitably be some degree of variability in the absolute bound (incorporated) fractions between individual cell lines depending on expression level and cell cycle profiles (Ref 69, Hansen et al used U2OS cells, Ref 71, Kimura and Cook used HeLa cells). In the context of our experiments, we would like to stress that the absolute bound fraction of H2B-HaloTag is only used as a very rough comparator in our initial characterisations to make the point that PRC1 does not appear to be highly bound to chromatin (Figure 2C). Additionally, the primary purpose of H2B-HaloTag was to act as a protein that would stably incorporate into chromatin and function as a photobleaching control for subsequent 2 Hz and 0.033 Hz residence time imaging. In this context, the absolute bound fraction of H2B-HaloTag is not relevant, as only molecules incorporated into chromatin are measured. As such, the absolute incorporated fraction of H2B-HaloTag will not influence our conclusions about Polycomb system dynamics or chromatin binding.

Nevertheless, to further characterise and validate the H2B-HaloTag cell line we have carried out new FRAP analysis (Figure S2C). Importantly, the FRAP recovery curves were consistent with a predominant, but not complete, incorporation of H2B-HaloTag into chromatin. Furthermore, in line with our SPT measurements, we observe a larger unbound (unincorporated) fraction of H2B-HaloTag in ESCs than was previously observed for H2B-GFP in HeLa cells (by Kimura and Cook, Ref 71). While our SPT and FRAP experiments for H2B-HaloTag are in good agreement, we are cautious in making any more quantitative comparisons between the two sets of measurements as recoveries in FRAP experiments can depend on a number of features relating to diffusion and binding, not simply the proportion of bound and free molecules⁵.

Finally, regarding the accuracy of our particle tracking, we would like to highlight that we were extremely careful in designing our experiments to ensure that the number of molecules photoactivated in 67 Hz SPT experiments was very low. This helped to ensure that we would not create overlapping tracks and that our tracking would be accurate. To reassure the reviewer, we have now included example videos to illustrate this point in the revised manuscript (Videos S1-3) and also drawn attention to the low level of photoactivation in the Methods section. Therefore, we believe that our SPT experiments are accurately tracked, carefully controlled, and highly rigorous.

2. How many particles of RING1B-HT (in a single cell) were analyzed? The number of particles analyzed and typical video data for each experiment would be very useful for a better understanding.

As suggested by the reviewer we have now included a table (new Table S2) indicating how many particles (molecules) were analysed on average for each cell line in 67, 2, and 0.033 Hz tracking experiments. Additionally, we have also provided examples of typical input videos for each type of tracking experiment (Videos S1-5). We hope this aids the reviewer and readers in better understanding the nature of these imaging data and the conclusions that we draw from their analysis.

3. Line 171, "Polycomb bodies accounted for just 1.3% of the total nuclear volume..." Correct volume estimation of Polycomb bodies by fluorescence may be difficult because of the optical resolution limitation unless their body sizes are large enough.

To segment Polycomb bodies we set a lower volume limit of $0.029 \mu\text{m}^3$ which corresponds to a diameter of 380 nm for a spherical object of this volume (lines 897-899). Objects of this diameter can be resolved in the X and Y-axis on our spinning disk confocal microscope. Although this falls below the Z-axis resolution of our microscope, accurate segmentation in Z will be aided by the deconvolution approach we have used. It is possible that smaller accumulations of Polycomb proteins might exist and that these could only be identified using super-resolution microscopy. However, our goal in this analysis was to quantify the properties of Polycomb bodies, which are generally considered to be the large nuclear foci containing Polycomb proteins that are evident in conventional fluorescence microscopy⁶⁻⁹. Therefore, we believe that our analysis accurately measures the volume of Polycomb bodies and allows for general comparisons to other cellular measures, including the volume of the nucleus.

4. Line 274, "We estimate that on average each Polycomb body has 9 PCGF2 and 10 RING1B molecules..." I wondered how the authors estimated the average numbers: 10 RING1B molecules for each Polycomb body.

To determine the mean number of RING1B molecules per Polycomb body we first measured the total fluorescence signal in the nucleus and then determined the fluorescence equivalent to a single molecule of RING1B, based on our estimate of the number of RING1B molecules

per cell (Figure 5C). We next calculated the fraction of that signal arising from Polycomb bodies. Finally, we used the number of Polycomb bodies per nucleus to determine the fraction of signal arising from an average Polycomb body, and, therefore, how many molecules this corresponded to. The same approach was used for PCGF2, CBX7 and PCGF6. We have now provided a more detailed description of this calculation in the Methods section (lines 907-911).

5. Fig. 2I and Fig. S4B. While the SPT demonstrated that 20% of RING1B was bound to chromatin, the FRAP data showed that ~40% of RING1B bound to chromatin stably in Polycomb bodies. It would be nice to see the RING1B behaviors in Polycomb bodies by SPT

We agree that SPT for RING1B specifically in Polycomb bodies would be interesting and provide more information about RING1B behaviour in these sites. In fact, we have thought extensively about the feasibility of such experiments. However, Polycomb bodies make up only a small fraction of the nucleus. Given that accurate SPT relies on random and sparse photoactivation of RING1B molecules in the nucleus, capturing molecules that by chance engage with a Polycomb body would require us to extend our acquisition times by orders of magnitude. Because Polycomb bodies diffuse in the nucleus and ESCs are not stationary, their location would change significantly over the course of these extended SPT experiments. Therefore, we would require an independent fluorescent marker for Polycomb bodies that could be contemporaneously imaged in a separate channel in order to determine which SPT tracks corresponded to the location of a Polycomb body at each time point. Unfortunately, this is not possible on our TIRF microscope where we can only image one fluorescent channel per video. We hope to acquire new hardware and develop such approaches for future studies.

and also the RING1B localization by STORM.

As suggested by the reviewer we have now carried out a STORM reconstruction from an extended 67 Hz exposure SPT experiment for RING1B-HaloTag (Reviewer Figure 1). We observe foci in the STORM reconstructions that are of approximately similar size to those seen in spinning disk confocal images in the same cell line, suggesting these might correspond to Polycomb bodies. Characterising the tracks which localised to these regions using Spot-On revealed that >95% of these tracks arose from chromatin bound molecules, in fitting with the large fraction of non-recovering signal we observe in FRAP of Polycomb bodies. However, to ensure that these localisations correspond to Polycomb bodies, we would require an independent fluorescent marker of Polycomb bodies that could be imaged contemporaneously. As described above, this is not possible on our TIRF microscope, where we can only image one fluorescent channel per video.

Reviewer Figure 1. STORM reconstruction of RING1B localisation in a single nucleus.

A STORM reconstruction using the mean X and Y positions for each RING1B molecule from a 5-minute 67 Hz SPT experiment binned to each pixel for a single nucleus. The key indicates the shade corresponding to the number of localisations within a pixel. Scale bar = 1 μm .

I wondered whether Polycomb bodies might have more stably bound PRC1 than the authors estimated.

It is possible that our FRAP experiments might underestimate the stability of the bound fraction in Polycomb bodies if there were slight movements in the location of the Polycomb body during the course of our live-cell imaging experiments (as described above). We have now highlighted this limitation in the revised manuscript on lines 174-178.

6. Fig. 8B-E. The rapid depletion experiments using dTAG system is very nice, but dTAG-13 treatment for 96 hrs seems to be too long because indirect effects might be negligible. In addition, their differences (with and without dTAG-13) on the bound fractions are not so convincing (Fig. 8C and D).

We chose to deplete PRC2 for 96 hours as this is when we achieved maximal loss of H3K27me3. At this time point we do not anticipate that any significant indirect cellular effects will have occurred that could complicate our measurements as PRC2 removal is known to have very little effect on transcription or ESC cell viability, even after long term culture in its absence¹⁰⁻¹³. Importantly, we find that PRC2/H3K27me3 depletion has very specific effects on the binding of the canonical PRC1 component PCGF2, which is known to interact with H3K27me3, whereas the variant specific PRC1 component PCGF6, which does not interact with H3K27me3, is unaffected. We agree with the reviewer that the effects on PCGF2 binding after PRC2/H3K27me3 depletion are modest, and believe this is consistent with other binding activities contributing significantly to its interaction with chromatin. This is also in agreement with our observation that there is only a modest effect on PCGF2 localisation to Polycomb bodies after PRC2/H3K27me3 depletion (Figure 8F – H). We have now drawn attention to this important finding on lines 391-400 of the revised manuscript.

Minor comments:

1. Based on subunit composition, PRC1 complexes are grouped into canonical and variant forms. For general readers, it might be helpful to have some schemes for them.

We thank the reviewer for this helpful suggestion. We have now included schematics for PRC1 complexes in Figures 1A and 5A that will be a helpful aid for generalists.

2. Line 152, "38% of observed binding events were stable..." I wondered where the value came from?

The percentage of observed binding events which are stable is derived from the biexponential decay function fitted to the 1-CDF curve from residence imaging (2 Hz and 0.033 Hz) experiments. We have now described in more detail the general decay function used in the 'Single molecule binding time analysis' section of the Methods (lines 840-845). Briefly, this biexponential decay consists of a rapidly decaying fraction and a slowly decaying fraction. The fraction of observed binding events which are stable is therefore derived from the contribution to the overall decay from the slowly decaying fraction (if A represents the fast decaying fraction, then this is $1 - A$). We have altered the text where this is first explained to better describe our approach and the resulting measurements (lines 146-155 and 157-160).

3. Line 190, "To estimate the maximum number of molecules that could possibly bind to RING1B-occupied target sites identified in ChIP-seq (18,643 sites)..." Since ChIP-seq data normally come from many cells, it is possible that actual RING1B-occupied target sites in the single-cell might be much fewer.

As the reviewer notes, ChIP-seq quantifications represent a population average of binding events from millions of cells. Current ChIP technology does not enable single cell/single loci binding quantifications. Therefore, in individual cells the actual number of bound RING1B molecules at a given site could be more or less than our estimations. However, RING1B ChIP-seq distributions provide a reasonable ensemble estimate of the relative occupancy across binding deciles, and the RING1B bound sites used in this analysis have been extensively validated as true binding sites based on quantitative ChIP-seq analysis comparing wild type and RING1B knockout cells¹⁴. Furthermore, when PRC1 is removed, many of these sites are associated with genes which are subject to derepression, indicating they are under the control of PRC1 activity^{10,14,15}. Even if we were to assume (incorrectly) the most extreme possibility in which all measured RING1B binding events were concentrated into the top decile of RING1B bound sites (1864 sites), the density of RING1B binding would still only be 0.44 molecules per kilobase, and thus many sites would still lack a bound RING1B molecule. Even under these unrealistic conditions it is therefore difficult to imagine how PRC1 complex occupancy could explain repression, further supporting our conclusion that H2AK119ub1, which is much more numerous in these sites (lines 492-510), is most likely to be the central determinant of gene repression by PRC1. Nevertheless, at all stages in the manuscript we draw attention to the fact that our occupancy levels are estimates and make it clear they are based on a series of careful quantitative measurements and some reasonable, but not absolute, assumptions. This is to ensure that the reader is aware that we are not making direct in situ single-cell single-locus measurements, which are of course not yet technically feasible.

4. Line 322. The sentence has no period.

We have now corrected this oversight.

5. Fig. S1A. What does Panel A show? I guessed displacement histograms. A proper explanation of Spot-On analysis is needed so that the readers do not have to go to Ref. 69.

Panel A shows displacement histograms containing the first four steps from all tracks starting from the indicated Δt . Curves representing two- and three-state model fits are overlaid on these histograms. We have now expanded the Materials and Methods section to provide more detailed information describing the SPT analysis and a more detailed explanation of the Spot-On analysis method. We hope this ensures that the reader does not need to go to Ref 69 to appreciate the approaches we have employed.

Figure legends for A and B are in reverse order?

We thank the reviewer for noticing this error. We have now fixed the figure legend in question.

6. Fig. S2A Legend. What was alkaline phosphatase (AP) staining for? A differentiation marker? A proper explanation is required for general readers.

Expression of alkaline phosphatase (AP) is elevated in pluripotent embryonic stem cells and diminishes as cells differentiate. In AP staining experiments, cells are treated with a substrate on which AP acts, staining the cells pinkish-red if they have high levels of AP. As such, pluripotent cells stain strongly and differentiated cells more weakly. Therefore, we use this assay as a proxy for maintenance of pluripotency in experiments where we remove endogenous PRC1 and rescue with exogenously expressed RING1B or RING1B^{NBM}. We have now highlighted this in the text of the revised manuscript to make clearer the purpose of this experiment (lines 220-225 and 561-563).

7. Fig.8B. The H3 bands are too faint.

We thank the reviewer for pointing this out. We have now replaced the histone H3 western blot in Figure 8B with a longer exposure.

Reviewer #2 (Remarks to the Author):

Polycomb group proteins are important for the maintenance of cellular identity. However, how they are targeted to chromatin and how long they spend there is not fully understood. The authors investigate the dynamics of Polycomb Repressive Complex 1 (PRC1) using single particle tracking and report that only a small fraction is chromatin bound. They take on the interesting but challenging task of estimating the number of PRC1 molecules found at bound chromatin sites. There are numerous different 'flavours' of PRC1 and the authors deliver an impressive effort in dissecting the dynamics of a variety of these 'flavours'. The tagging of endogenous alleles with HaloTags/dTags is elegant and should be commended. The manuscript is very well written and while some of the findings are based on assumptions/estimations, they are nonetheless thought-provoking and will be of value to the Polycomb field.

We thank the reviewer for their positive comments describing the elegant nature of our tagging approach and for indicating the value of our findings to the field.

Major points:

1. Given that some PCGF2 can incorporate into variant PRC1 complexes (Tavares et al. 2012; Gao et al. 2012), the authors could have extended their analysis in Figure 6C and D as well as S4 to include CBX7, which would allow them better focus on canonical PRC1.

As suggested by the reviewer, we have now included analysis for CBX7 in Figures 6C and D, and S5. Using ChIP-seq data we find that CBX7 has a very similar distribution to H3K27me3 and PCGF2, and these enrichments correspond to sites with higher levels of RING1B (lines 301-310, Figure 6C and D). Furthermore, we have carried out new FRAP experiments for CBX7 and analysed its enrichment in Polycomb bodies (lines 284-297, Figure S5). We find that CBX7 exhibits the same slow, incomplete recovery within Polycomb bodies as RING1B and PCGF2, albeit with slightly higher overall recovery. CBX7 is less enriched (1.7-fold) in Polycomb bodies than PCGF2 (2.3-fold), but more enriched than RING1B (1.3-fold). This further demonstrates that canonical PRC1 subunits are more enriched in Polycomb bodies.

2. If possible, the authors could repeat the analyses in Figure S4 for RYBP to see if there is a differential enrichment of canonical and variant PRC1 at Polycomb bodies. The fact that PCGF2 is more enriched at Polycomb bodies than RING1B (Figure 2I and J) suggests that there will be a difference between canonical and variant PRC1 enrichment. However, CBX7 (canonical PRC1) and RYBP (variant PRC1) enrichment at Polycomb bodies would be a more suitable comparison to address this question.

We agree with the reviewer that it would be useful to directly compare variant and canonical PRC1 enrichment in Polycomb bodies using our HaloTag fusions of CBX7 and RYBP. However, as is evident from fluorescent images (Figure 5B and Reviewer Figure 2), RYBP does not appear to be significantly enriched in Polycomb bodies. In fact, we observe fewer than 10 bright nuclear foci of RYBP per cell. As such, we cannot carry out efficient Polycomb body segmentation and enrichment analysis. However, we were able to identify and segment Polycomb bodies for the variant PRC1 subunit PCGF6. In line with the reviewer's prediction, PCGF6 was less enriched (1.2-fold) than CBX7 (1.7-fold) (lines 284-287 and 325-334). Together these new analyses further support the contention that canonical PRC1 complexes are more enriched in Polycomb bodies.

Reviewer Figure 2. Example deconvolved images for RYBP, PCGF6 and PCGF2.

Example Z-slices from deconvolved Z-stacks for Polycomb body segmentation for RYBP, PCGF6 and PCGF2. Scale bar = 10 μ m.

3. In Figure 8, dTAG-SUZ12 could have been introduced into the cell line expressing HaloTagged CBX7 to focus on the contribution of H3K27me3 to canonical PRC1 dynamics exclusively.

We agree with the reviewer that examining CBX7 dynamics in the absence of H3K27me3 is important. However, the dynamics of CBX7, and other CBX proteins that are incorporated into canonical PRC1 complexes, have previously been investigated in detail in mESCs¹⁶. This

demonstrated that individual CBX proteins have different dependencies on H3K27me3 for chromatin binding, with CBX7 binding being more reliant on H3K27me3, whereas CBX2, CBX4 and CBX8 were less dependent. However, this study did not examine the net effect that these diverse CBX proteins would impart on canonical PRC1 behaviour. Given that multiple CBX proteins are expressed in ESCs and can be incorporated into canonical PRC1 complexes, here we sought to investigate more generally how H3K27me3 influences the binding and localisation of canonical PRC1. To achieve this we used PCGF2 as a proxy for canonical PRC1 complexes as it is highly expressed in ESCs. Interestingly, we find that removal of PRC2 and loss of H3K27me3 has only a modest effect on the binding of PCGF2 to chromatin and its localisation to Polycomb bodies. This suggests that PCGF2-containing PRC1 complexes can continue to interact stably with chromatin through H3K27me3-independent mechanisms (lines 400-405).

Minor points:

1. In Figure 1C and 5B, if available, the authors could include a z-slice from WT cells (lacking HaloTag fusions) treated with JF549 as negative controls to highlight the specificity of the staining in the HaloTag fusions.

We have now included a Z-slice from live WT cells labelled with JF₅₄₉ in Figure 1D as suggested by the reviewer. This nicely demonstrates the specificity of JF₅₄₉ for cells expressing RING1B-HaloTag.

2. The authors do western blotting to show that they have successfully HaloTagged their proteins of interest (Figure 1A and S3). For the benefit of those who may not be very familiar with Halo-Tags, they might mention the expected size of the tag is 33kDa in legend of Figure 1A so that readers can appreciate that the expected size shift takes place.

We thank the reviewer for pointing out this oversight. We have now indicated the size of the HaloTag and linker which was added to the protein in the figure legends for Figures 1A and S4.

For some of the Halo-tagged proteins, there are some unfortunately sized background bands (i.e. for PCGF2 and PCGF3). In the Methods section, the authors write that, "The integrity of homozygote clones was then further validated by sequencing the PCR products to ensure the expected HDR event had occurred". If available, perhaps some of this sequencing data could be included as supplemental data.

As suggested by the reviewer, we now show the PCR products across the insertion site in untargeted and PCGF2-HaloTag or HaloTag-PCGF3 cell lines. This illustrates a band shift that corresponds to the insertion of the HaloTag sequence (Figure S4G). We have also included sequencing traces from either end of the insertion site for these cell lines (Figure S4H). Additionally, we have now carried out immunoprecipitation of RING1B in each of the cell lines in Figure S4A – F and carried out western blot analysis for the HaloTagged protein. This illustrates that each HaloTag protein is incorporated into PRC1 and also eliminates the background bands that in some instances migrated at similar positions to the tagged protein.

3. The authors estimate that, "there would be on average 0.1 RING1B molecules for every kilobase of RING1B-enriched chromatin." They also estimate that, "sites in the top decile of RING1B density would still have fewer than 0.3 molecules per kb" and furthermore, they report that these are overestimations. These estimates are surprisingly low. Could the authors please comment on whether there are any technical limitations or caveats to the protocols used which could result in an underestimation of the amount of RING1B molecules on chromatin.

We agree that the estimated number of molecules per site is low and this is a very important new observation that emerges from our PRC1 complex quantification and chromatin binding measurements. This observation was also surprising to us and therefore we have extensively considered whether there are any technical or analysis limitations that could lead to an underestimation of binding at target sites. For example, if RING1B-HT was functionally defective this could somehow lead to underestimation of binding in our SPT experiments. However, we find that RING1B-HT is expressed at the same level as endogenous RING1B (Figure 1B and S1A), forms PRC1 complexes normally (Figures 1C and S1B and C), and can rescue the viability of RING1B knockout cells (Figure S3A). Therefore, we believe that RING1B-HT is a good proxy for endogenous PRC1 binding. In addition, its binding characteristics as measured by SPT are consistent with other PRC1 complex components (Figures 6 and 7). Furthermore, in our calculations of density, we purposely made a series of conservative assumptions to ensure they were much more likely to be overestimates than underestimates. For example, we assume that only two copies of each binding site exist in each cell (i.e. that the average ESC is $2n$), when, in reality, the average ESC likely has closer to three copies of each binding site (see, for example, Cattoglio et al., 2019). Furthermore, we assumed that RING1B-containing complexes bind exclusively to chromatin in RING1B ChIP-seq peaks (which make up only 1.6% of the genome), despite evidence that PRC1 can also deposit H2AK119ub1 elsewhere in the genome at low levels^{10,17}. Therefore, based on our rigorous characterisation of the RING1B-HT line and the conservative assumptions inherent to our calculations, we believe that it is highly unlikely that the values we report are an underestimation.

4. In Figure 4, 6, 7 and 8 – how exactly are ‘bound’ and ‘stably bound’ molecules defined? Perhaps the authors could make this distinction more obvious in the text.

We apologise for the confusion that has arisen from the explanation of these two measurements. The proportion of bound molecules is determined from 67 Hz SPT experiments and Spot-On analysis and represents the proportion of all molecules predicted to be bound to chromatin. The fraction of stably bound molecules is derived from the contribution of the slowly decaying component of the biexponential fit to the survival of tracks in 0.5 Hz and 0.033 Hz experiments. Stably bound molecules are therefore a subset of bound molecules which exhibit long binding times. We have now altered how these measurements are described in the text to make the differences between these two approaches and how they are derived clearer (lines 146-155 and 157-160).

5. In the text, the authors write that 3.6% of PCGF2 is found in Polycomb bodies. I’m confused as to how this figure was calculated. Earlier in the manuscript they write “our measurements revealed that Polycomb bodies accounted for just 1.3% of the total nuclear volume and that RING1B fluorescence signal inside Polycomb bodies was only 1.3-fold more intense than the surrounding nucleus (Figure 2J). This means that only 1.7% of total PRC1 signal originates from Polycomb bodies”. This suggested to me that 1.3×1.3 was used to reach the 1.7% value. S4A suggests that PCGF2 signal is ~2.3 more intense in Polycomb bodies and 2.3×1.3 would equal to 3% (rather than 3.6%). Apologies, if I’ve misinterpreted something here.

We thank the reviewer for pointing this out. This difference arises because the percentage of nuclear volume taken up by Polycomb bodies differs between RING1B and PCGF2. While Polycomb bodies make up 1.3% of the nuclear volume for RING1B, for PCGF2 they make up 1.6%. This is likely due to better segmentation of Polycomb bodies using signal from PCGF2, where signal from the rest of the nucleus is lower, which is reflected in the higher enrichment for PCGF2. We would like to note that the percentages of the nuclear volume and total signal

are derived directly from the measurement of segmented Polycomb bodies and nuclei. Enrichment is calculated separately by normalising the average Polycomb body/non-Polycomb body fluorescence signal for each nucleus to the median non-Polycomb body fluorescence. Therefore, while enrichment is influenced by how well Polycomb bodies are segmented, it is not directly dependent on the calculated percentage of the nucleus made up by Polycomb bodies.

6. The authors estimate that PCGF3/5-PRC1 complexes deposit a H2AK119ub modification every 16 seconds in order to maintain H2AK119ub levels. A prior single particle tracking study reported a “target search time” for CBX7 (Tatavosian et al. 2018). With the data the authors have collected, would it be possible to calculate a target search time for PCGF3/5? If possible, this could potentially support their 16 second estimation.

As suggested by the reviewer, we have now determined the search times for PCGF3. To achieve this we calculated the search time for both long (39 s) and short (1.5 s) binding events as both could lead to deposition of H2AK119ub1. The search time for more rare long binding events (3.5% of all PCGF3 molecules) was 1070 s. However, the search time for the much more common short binding events was only 15 s, in line with the estimated frequency that would be required for pervasive deposition of H2AK119ub1 by PCGF3/5. We thank the reviewer for suggesting this analysis as it further highlights how the dynamics of these complexes are linked to their catalytic activity in vivo. We have added a description of this new result to the main text (lines 354-359), and a description of how the calculations were carried out in the Methods section of the revised manuscript.

Reviewer #3 (Remarks to the Author):

This paper examines Polycomb repressive complex 1 (PRC-1), a well-known chromatin-interacting transcriptional repressor. The authors used CRISPR/Cas9 to genetically engineer a stable cell line integrating a HaloTag into the Ring1b gene, a component of the core subunit of PRC1 canonical and variant complexes. Single-particle-tracking was performed on RING1B-HaloTag, revealing 3 diffusing populations of PRC1. Using H2B-Halo as a control, only a small amount of PRC-1 complex is determined to be chromatin bound. A mathematical approach in comparison with ChIP data is used to estimate PRC-1 occupancy, demonstrating low occupant density at PRC-1 target sites. The authors track a non-functional mutated RING1B, and quantification of bound molecules is similar to wild-type protein. The authors conclude that interactions between the catalytic core of PRC1 and the nucleosome does not contribute significantly to chromatin binding and instead propose auxiliary proteins are responsible.

The authors continue by CRISPR-editing HaloTags into known proteins in variant PRC-1 complexes. Fluorescence quantification reveals various levels of expression, with PCGF2-PRC1 being the most abundant. SPT of variant complexes revealed distinct dynamics of diffusion and binding behavior. A dTAG system was then used to interrupt the placement of H3K27me3 across the genome. H3K27Me3 reduction did not have an effect on binding stability, but did slightly reduce the bound fraction of RING1B and PCGF2. Finally, SPT examination of the binding behavior of N- and C- terminal fragments of PCGF-1 demonstrated that the C-terminal fragment is responsible for binding behavior.

Overall, this is a nice paper demonstrating the quantitative power of single particle tracking when combined with CRISPR genome editing and ChIP-Seq. An impressive number of genes were CRISPR'd and the resulting SPT data are of high quality and rigor. Since the authors quantify such a low target-site occupancy for PRC1, especially the PCGF3/5-PRC1 variants responsible for most H2AK119ub1 deposition and gene repression, they conclude that

H2AK119ub1 must be deposited in a hit-and-run like fashion (they estimate that each of the ~6000 PCGF3/5-PRC1 complexes deposits one H2AK119ub1 mark every ~16 seconds). This suggests the H2AK119ub1 modification itself, rather than direct occupancy by PRC1, accounts for most of the PRC1-mediated gene repression in ESCs.

We thank the reviewer for their supportive comments and drawing attention to the high quality and rigour of our SPT experiments and the new insight that emerges from these approaches.

Major Points

1. How much of RING1B is in complex? The authors use RING1B as a proxy for PRC-1 behavior, and demonstrate that HaloTag does not disrupt normal complex formation. However, if a significant amount of RING1B exists outside of PRC-1 complex, then analysis of RING1B SPT behavior would not provide an accurate representation of PRC-1 behavior. The authors should clarify this issues and tone down the text to make it clear that the dynamic they measure is a combination of subunit assembly kinetics and binding kinetics.

Expanding on the above – an equally valid interpretation is that the majority of RING1B is not in complex, but in complex, the bound fraction is much higher. That is, the majority of observed bound fraction is actually when RING1B is in complex. Other data within the paper also alludes to the fact that complex formation is more important than individual binding (RING1BNBM mutant binding is reduced by a small but significant amount, and mutant which does not interact with PRC1 auxiliary factors has same binding as negative control).

We thank the reviewer for bringing up these extremely important points and fully agree that if significant amounts of RING1B were not incorporated into PRC1 complexes or if the HaloTag affected PRC1 complex formation, this could affect our interpretation of subunit dynamics in the context of their respective complexes. Therefore, to investigate this point in more detail, we have now carried out extensive new biochemical characterisation of RING1B and PRC1 complexes in wild type and HaloTagged cell lines as follows:

(1) Firstly, we have used size exclusion chromatography to fractionate cellular proteins/protein complexes from nuclear extracts. Using this approach, we compared native RING1B and RING1B-HT directly to non-complexed monomeric RING1B and RING1B-HaloTag that we have recombinantly expressed and purified from E. coli (Figure S1B and C and lines 115-118). Importantly, this demonstrated that RING1B and RING1B-HT are found predominantly in high molecular weight fractions that are distinct from monomeric recombinant RING1B or RING1B-HT. Furthermore, the elution profiles of RING1B and RING1B-HT were very similar, with the exception that the RING1B-HT protein complexes fractionated at a slightly larger sizes due to the inclusion of a HaloTag. Together, these observations demonstrate that RING1B/RING1B-HT are predominantly found in large molecular weight protein complexes and that addition of a HaloTag to RING1B does not affect PRC1 complex formation.

(2) To further corroborate this point we have also carried out western blot analysis of variant PRC1 (PCGF1/3/5/6 and RYBP) and canonical PRC1 (CBX7 and PCGF2) complex components in fractionated nuclear extracts (Figure S1B and C). Importantly, this demonstrates that PRC1 complex components migrate predominantly in large molecular weight fractions with RING1B and RING1B-HT, providing further evidence that RING1B-HT forms PRC1 complexes appropriately and predominantly resides in PRC1 complexes. This is in agreement with RING1B immunoprecipitations from RING1B-HT lines that also demonstrate appropriate biochemical interactions with vPRC1 and cPRC1 complex components (Figure 1C).

(3) *In addition to these detailed characterisations of RING1B-HT PRC1 complexes, we have now also validated that other HaloTag PRC1 components are appropriately incorporated into PRC1 complexes. To achieve this, we carried out RING1B immunoprecipitations from individual HaloTag cell lines and used western blot analysis to ensure the HaloTagged protein is efficiently incorporated into PRC1 complexes (Figure S4A – F).*

Together these extensive new biochemical analyses further validate the use of our RING1B-HT line to characterise PRC1 complex behaviour, based on its role as a core structural component of PRC1 and almost complete incorporation into PRC1 complexes. More broadly, this analysis also confirms that HaloTag fusions of other PRC1 components, that we have used to dissect PRC1 complex dynamics and chromatin binding, form PRC1 complexes efficiently. We thank the reviewer for bringing up this important consideration and we believe these new validations significantly strengthen the manuscript.

2. Fig. 2F,G (related to Line 155 30 s interval distribution): *Could it be that these are spots where more than one RING1B molecule came in and out? Is the density of fluorescent molecules at any given time low enough to rule this out?*

In order to avoid this important issue, in 2 Hz and 0.033 Hz residence time imaging experiments the density of photoactivated molecules was kept very low (please refer to example residence videos, Videos S4 and S5) and there was no further photoactivation after acquisition was initiated. This makes it highly unlikely that two (or more) molecules would overlap sufficiently to be tracked as a single molecule. Furthermore, because H2B is incorporated into chromatin, in parallel experiments we examined H2B-HaloTag localisations under the same imaging conditions and used this to limit the dimensions of individual localisations for RING1B-HT (and other HaloTag protein localisations) and filter out localisations that could possibly have arisen from more than one overlapping molecule. Based on these experimental considerations we believe we are effectively capturing the behaviour of single molecules.

To rule out the possibility that long binding events are due to repeat binding, it would be good to show the intensity through time. The intensity should remain constant and photobleach in one step at the end of the measurement.

As suggested by the reviewer, we have now plotted normalised intensity over time for a RING1B-HT 2 Hz and 0.033 Hz residence time imaging experiment (Reviewer Figure 3). These plots indicate that the intensity of each molecule remains approximately constant throughout the experiment. More variability is observed in 0.033 Hz experiments, but this is likely due to the increased propensity for chromatin diffusion and cell movement between frames which can cause a change in the molecule's position in the focal plane, and therefore the intensity of the signal detected. Again, we believe these quantifications further support our contention that residence time imaging captures the behaviour of single molecules.

Reviewer Figure 3. In 2 Hz and 0.033 Hz residence imaging experiments fluorescence intensities of tracked molecules remain approximately constant.

Plots of intensities of tracked RING1B molecules from example 2 Hz and 0.033 Hz residence experiments. Intensities for each molecule are normalised to the mean of all intensities for that molecule.

Minor Points

2. Full, uncropped gels for examination? I have no reason to doubt the veracity of the authors gels, but it is good practice that all uncropped gels images are provided for examination.

We have now included a supplementary file containing all uncropped gels used in the manuscript as suggested by the reviewer (Source Data File).

3. Fig. 2H – Is this a maximum intensity projection?

We apologise for the confusion here. The right hand panel of Figure 2H is a maximum intensity projection of a RING1B-HaloTag nucleus labelled with Halo-JF₅₄₉. We have now adjusted the labelling and figure legend to make this clearer.

4. Line 359-360 – Full loss of H3K27me3 is not what their data shows, but a large reduction does seem to be observed.

As suggested by the reviewer we have adjusted the text to reflect that a small quantity of H3K27me3 remains (lines 386-388).

Reviewer #4 (Remarks to the Author):

In this paper, the authors have endogenously tagged in mice embryonic stem cells different proteins that form the PRC1 complex. Using a HaloTag, the expression level, localization and dynamics of these proteins have been characterized. Notably, the authors have performed single protein tracking to monitor the fraction of proteins that strongly associates to the chromatin versus the fraction that freely diffuses in the nucleus. Using different mutations or protein depletion strategies, they have attempted to perturb the association of the protein of

interest with the chromatin in order to understand the mechanisms of binding of the different members of the complex. While the manuscript is clearly written and presents interesting and surprising insights on the highly dynamic nature of the PRC1 complex, I have a few comments that I hope the authors can address.

We appreciate the reviewer's supportive comments and for noting that our findings provided interesting and surprising insights. We have addressed the reviewer's comments below.

The authors have used HaloTag-H2B and HaloTag-NLS as control for their SPT experiments. However, with the H2B control, they observe 60% of bound molecules, while FRAP data in the literature observe 90% of stably bound H2B (for instance: Kimura JCB 2001). Conversely, for a freely diffusing HaloTag-NLS, 11% of bound protein is measured. Where do those discrepancies come from? I would suggest performing additional experiments with control cells bearing no HaloTag to check if the Halo dye can interact non-specifically with some proteins or whether the washing step leaves unbound dye in the cells. FRAP experiments comparing HaloTag vs GFP tagged H2B or NLS could also help clarify this problem.

We thank the reviewer for bringing up these important points. Endogenous histones are expressed and incorporated into chromatin almost exclusively during S-phase¹⁻⁴. Unlike endogenous histones, H2B-HaloTag is expressed from a non-histone promoter and will be produced throughout the cell cycle. H2B-HaloTag expressed outside of S-phase will not be immediately integrated into chromatin and its expression level can influence the overall unbound (non-integrated) fraction. Therefore, depending on the exogenous expression approach used, there will inevitably be some degree of variability in the absolute bound (incorporated) fractions between individual cell lines depending on expression level and cell type specific cell cycle profiles (Kimura and Cook 2001 used HeLa cells). We believe these technical considerations likely explain the relative differences in the absolute levels of bound exogenously expressed H2B in differing experimental configurations. Nevertheless, as suggested by the reviewer, we have further characterised and validated our H2B-HaloTag cell lines by carrying out FRAP analysis for H2B-HaloTag (and RING1B-HaloTag as a comparison) (Figure S2C). Consistent with our SPT measures, we observed that there was a larger unbound fraction of our H2B-HaloTag protein in ESCs than was observed previously for H2B-GFP in HeLa and U2OS cells^{18,19}. Direct comparisons between SPT and FRAP experiments are challenging because recovery in FRAP experiments can depend on a number of features relating to diffusion and binding of molecules and not simply the proportion of bound and free molecules⁵, so we are cautious in making any more detailed comparisons of these SPT and FRAP measurements in this context.

In the case of HaloTag-3xNLS, it has been previously documented that proteins bearing an NLS sequence exhibit non-specific binding via the positively charged NLS to negatively charged DNA²⁰. We observe a bound fraction consistent with the measured bound fraction for HaloTag-3xNLS in the original Spot-On paper¹⁹. As suggested by the reviewer, we have now carried out new FRAP analysis of the HaloTag-3xNLS and this displays rapid and complete fluorescence recovery as expected for a protein undergoing brief, nonspecific binding events (Figure S2C).

In order to address whether unbound dye remains in cells after labelling, we have now quantified fluorescence signal from wild type and RING1B-HaloTag nuclei labelled with JF₅₄₉ (Reviewer Figure 4). Importantly, this revealed that background dye is unlikely to contribute significantly to our measurements given that the signal from wild type (untagged) cells was less than 3% of that in RING1B-HaloTag cells. Even this small amount of background is likely an overestimation because a proportion of the measured signal in wild type cells will arise due to cellular autofluorescence, which will not contribute to our SPT measurements. Furthermore,

when PA-JF₅₄₉ is conjugated to the HaloTag, its photochemical properties are improved²¹, meaning we will almost exclusively capture these events in SPT experiments. Therefore, based on the extremely low background signal in untagged cells, and the properties of the PA-JF₅₄₉, we believe any small amount of unbound dye will have a negligible effect in our SPT experiments.

Reviewer Figure 4. Wild type (untagged) cells exhibit low fluorescence signal after labelling with JF₅₄₉.

A box plot illustrating the relative JF₅₄₉ fluorescence signal measured for wild type (WT) and RING1B-HaloTag nuclei. Measurements were taken for 20 nuclei across 2 biological replicates.

The authors have used very low imaging frequencies to monitor the stable binding events of a single molecule to the chromatin. How can the authors verify that what they observe represents the persistent binding of the same molecule and is not the exchange of multiple freely diffusing species on a stable structure?

In order to avoid this important issue, in 2 Hz and 0.033 Hz residence time imaging experiments the density of photoactivated molecules was kept very low (please refer to example residence videos, Videos S4 and S5) and there was no further photoactivation after acquisition was initiated. This makes it highly unlikely that two (or more) molecules would overlap sufficiently to be tracked as a single molecule. Furthermore, because H2B is incorporated into chromatin, in parallel experiments we examined H2B-HaloTag localisations under the same imaging conditions and used this to set x and y migration limits for the localisations of other HaloTag proteins. This allowed us to filter out any residence time measurements that could possibly have arisen from the behaviour of two spatially distinct molecules. Under these imaging conditions and strict filtering criteria, tracking the exchange of multiple diffusing molecules would require them to do so without motion blurring (over 0.5 s or 1 s exposures), within the permitted displacement radius, and despite the low proportion of molecules that are both labelled and photoactivated. Nevertheless, to further corroborate the fact that only single stably bound molecules are being tracked we have analysed the intensity of localisations within each track in example 2 Hz and 0.033 Hz imaging videos and plotted their intensity (Reviewer Figure 3). These plots indicate that the intensity of each molecule remains approximately constant throughout the experiment consistent with the imaging of a single molecule. There is slightly more variability in 0.033 Hz experiments, but this is likely due to the increased propensity for chromatin diffusion and cell movement between frames which can cause a change in the molecule's position in the focal plane, and therefore the intensity

of the signal detected. Again, we believe these quantifications further support our contention that our residence time imaging captures the behaviour of single molecules.

In most of the figures, the authors present their data as the mean and the SEM of three biological replicates. However, there is a large diversity in the single-particle traces measured and I wonder if there isn't a lot of interesting information that could be extracted from this variability.

We agree with the reviewer that there is likely much more information in the tracks that we have acquired. In particular, in many of the longer tracks, it is possible to observe state transitions within tracks (Reviewer Figure 5A). In an attempt to extract additional information from these state transitions, we also analysed tracks from each cell line, and under different treatment conditions, using vbSPT²². vbSPT fits states, and state transitions, to the data. By classifying sections of tracks into these states, one can determine the times spent by molecules in different states, and the probability of transitions between states. A problem we encountered with this approach was the tendency of vbSPT to fit a very large (>10) number of states to the data when not constrained. We therefore restricted the model to 3 diffusing states in an attempt to compare to the outputs from Spot-On. It is important to note that vbSPT, unlike Spot-On does not account for defocalisation bias, and therefore produces differing estimates of bound fractions. Our vbSPT analysis indicated that molecules typically transitioned between states in the order fast \leftrightarrow slow \leftrightarrow bound, and did not transition directly between bound and fast. It also enabled us to determine dwell times for the bound states of different proteins observed in 67 Hz SPT (Supplementary Figure 5B). However, at this stage we concluded that this information did not substantially alter or provide clarification to our interpretations so have not attempted to describe or include this more complicated analysis in the manuscript. We hope to develop and apply further analysis approaches to extract and study the information within individual tracks in future work.

Reviewer Figure 5. vbSPT analysis of SPT tracks.

- (A) An example of long (215 localisations, 3.3 s (blue), 95 localisations, 1.4 s (purple), 131 localisations, 2.0 s (pink)) RING1B-HT tracks which exhibit both diffusing and binding behaviours. The asterisks indicate the first localisation of each track. The scale bar is 500 nm.
- (B) A table of dwell times of different PRC1 subunits in the bound state as determined by vbSPT using a 3-state model.

In the experiment performed with the mutant of RINGB that cannot bind to nucleosomes, the endogenous RINGB is still present. Can the presence of the functional endogenous protein

affect the dynamics of RINGBNBM? I suggest that the authors try to measure the behavior of the mutant allele in absence of the endogenous protein.

We thank the reviewer for bringing up this important point. In fact, we considered this extensively at the experiment design stage and in the end reasoned that it was important to carry out these experiments in the presence of endogenous RING1B. This is because we, and others, have previously shown that cell lines lacking H2AK119ub1 also have major reductions in H3K27me3 and that reduction of these two histone modifications can affect PRC1 occupancy^{14,15}. This is because vPRC1(RYBP-H2AK119ub1) and cPRC1(CBXs-H3K27me3) rely in part on these histone modifications for chromatin binding. In our initial characterisation of the RING1B^{NBM}-HaloTag lines we found that removal of endogenous RING1B caused a near complete loss of H2AK119ub1 (in agreement with its requirement for catalysis^{23,24}). Therefore, in the absence of endogenous RING1B our measurements of RING1B^{NBM}-HaloTag behaviour by SPT would have been confounded by a series of additional effects on PRC1 binding that were not a direct consequence of mutating its nucleosome binding activity. Hence, we believe our current experimental configuration where RING1B^{NBM}-HaloTag is studied in the presence of endogenous RING1B is the most informative way to study this mutation. However, we have also drawn attention to the possibility that the endogenous protein could influence our interpretations in the revised manuscript (lines 226-234).

The tagging of the various members of the PRC1 complex have been performed at the N or C terminus. Is there a specific reason for these different tagging strategies and can it explain some of the differences measured between the PCGF proteins?

All proteins were tagged at the N-terminus by default unless features of the gene required a C-terminal tag. In the case of RING1B and CBX7, C-terminal tags were employed as there were multiple annotated splice isoforms for these genes which used different first coding exons but had common translation stop sites. In the case of PCGF2, the annotated Pcgf2 transcripts had alternatively spliced 5' UTRs. Therefore, we generated a C-terminally tagged PCGF2 protein to avoid disrupting splicing. For proteins that were N-terminally tagged, alternative splicing isoforms were either not relevant (PCGF1, PCGF3, RYBP), or alternative splicing isoforms shared a common first coding exon (PCGF6). To further validate these tagging strategies we have now performed new RING1B immunoprecipitation experiments in all HaloTag cell lines to confirm they are appropriately incorporated into PRC1 complexes (Figure S4A-F). Therefore, we do not have any evidence to suggest that tagging strategy would influence the behaviour of Polycomb system components or any of the differences observed in our experiments.

On line 274, the author claim that the FRAP dynamics recovered for PCGF2 are similar to the ones for RINGB. From a visual comparison of the two graphs, I would say that their behaviors are not similar. I don't think they have the same mobile fraction or that the kinetics of the fast recovering species are close to each other. A quantitative assessment of the kinetic parameters would be needed. If the FRAP curves are indeed different, it becomes difficult to explain why the SPT results between RINGB and PCGF2 are so similar.

We thank the reviewer for suggesting we examine and compare our FRAP measurements in more detail. To achieve this, we overlaid the Polycomb body FRAP curves for RING1B, PCGF2 and CBX7 (Reviewer Figure 6). Comparison of the recovery curves for all three proteins outside of Polycomb bodies, which accounts for the vast majority of molecules, suggest they exhibit very similar behaviours. Biexponential curve fitting to the data indicates all three proteins exhibit similar mobile fractions and recovery times, consistent with their similar bound fractions as measured by SPT (Reviewer Figure 6). These proteins do, however, differ in their recovery within Polycomb bodies. CBX7 (0.28 non-recovering fraction) recovers

more than PCGF2 (0.41 non-recovering fraction). This may indicate that PCGF2 complexes containing other CBX proteins are more stably bound within Polycomb bodies than those containing CBX7. Similarly, the lower fraction of RING1B which does not recover (0.36) relative to PCGF2 may result from the presence of a small proportion of non-PCGF2-containing PRC1 complexes present within Polycomb bodies which are not stably bound. Importantly, while RING1B, PCGF2 and CBX7 behave differently within Polycomb bodies, our estimates of the proportion of each protein present in Polycomb bodies indicates that this represents only a small fraction of the total protein, and therefore cannot be compared directly to SPT which measures dynamics of molecules within the nucleus indiscriminately.

Reviewer Figure 6. RING1B, PCGF2 and CBX7 exhibit subtle differences in behaviour within Polycomb bodies.

Left panel: FRAP recovery curves for regions containing a Polycomb body or elsewhere in the nucleus (Non-Polycomb body) for RING1B, CBX7 and RYBP. The recovered fraction was measured relative to initial fluorescence intensity and corrected using an unbleached region. The error bars denote SEM for 20, 31 and 40 cells, respectively, for each of Polycomb body and Non-Polycomb body regions across 2 biological replicates. Right panel: a table indicating the parameters of fitted biexponential recovery curves to the raw data.

The authors claim that the Polycomb bodies are mostly composed of canonical PRC1 complexes. However, their images also show an enrichment of PCGF1,3,6 in these structures. I would suggest measuring the enrichment of these proteins in these foci in order to compare the abundance of each protein in the Polycomb bodies. Dual-labeling of the cells could also provide very interesting insights, but is probably outside of the scope of this paper.

As suggested by the reviewer, we have now measured the enrichment of PCGF6 in Polycomb bodies (Figure S6). Unfortunately, the low enrichment of PCGF1 and PCGF3 in Polycomb bodies, combined with weaker signal due to their expression level, made it impossible to accurately segment Polycomb bodies and therefore quantify enrichment for these proteins. Nevertheless, as expected for a variant PRC1 subunit, PCGF6 exhibited lower enrichment (1.2-fold) in Polycomb bodies than that of canonical PRC1 subunits PCGF2 (2.3-fold) and CBX7 (1.7-fold) (Figures S6, S5A). We estimate, based on this measurement, that Polycomb bodies on average contain 1 – 2 molecules of PCGF6. We have now included this important new observation in lines 325-334 of the revised manuscript. We agree with the reviewer that dual labelling of subunits would add further insight into the colocalisation of different PRC1 complexes in the nucleus and permit analysis of proteins which are less enriched. However,

this would require extensive new cell line engineering and characterisation that we believe is beyond the scope of this current study.

In Figure 9B and C, the full length and N-terminal fragment of PCGF1 display the same bound fraction, but for the N-terminal fragment no stably bound complex can be observed. I would suggest plotting the 1-CDF curves for these traces in order to better document this phenomenon. Mechanistically, I'm not sure how this behavior can be explained because intuitively I would have expected that in absence of the stably bound fraction the overall bound percentage would decrease. Additional FRAP measurements could also strengthen this observation.

As suggested by the reviewer, we have now examined the 1-CDF curves for the exogenously expressed HaloTag-PCGF1 and N-terminal fragment (Reviewer Figure 7A). However, we would like to point out that these plots do not provide a complete view of the differences between these constructs. This is because the average number of bound molecules that could be tracked per video was much lower for the N-terminal fragment, averaging fewer than 10 molecules (and often fewer than 5), compared to >50 molecules for the other constructs (Table S2). As suggested by the reviewer we have now carried out spot FRAP analysis of HaloTag-PCGF1 and the N-terminal fragment. The N-terminal fragment recovered more rapidly and completely, consistent with the loss of stable binding observed in SPT (Reviewer Figure 7B). A small fraction of fluorescence signal did not recover for the N-terminal fragment, suggesting that some stable binding may still occur, but this is much less than for full length PCGF1. Given that the bound fraction remains the same despite highly abrogated stable binding, we envisage that the time between binding events must be reduced for N-terminal PCGF1. There are a number of possible explanations for this finding. For example, the reduced size of the complex may enable less stable interactions to occur more frequently, or, in the absence of the complex's typical mode of binding, a secondary mode of binding predominates that occurs more frequently but with lower stability. We have now drawn attention to this interesting result in lines 424-431 of the revised manuscript and discussed possible explanations.

Reviewer Figure 7. A PCGF1 N-terminal fragment exhibits reduced stable binding.

- (A)** Dwell time distributions (1 – cumulative distribution function, CDF) and fitted biexponential decay curves for immobile full length PCGF1 (FL) and PCGF1 1-139 molecules for a representative biological replicate of movies acquired in 2 Hz SPT experiments. $n = 2$ biological replicates of 8 videos each.
- (B)** FRAP recovery curves for full length PCGF1 and PCGF1 1-139. The recovered fraction was measured relative to initial fluorescence intensity and corrected using an unbleached region. The error bars denote SEM for 40 cells for each protein across 2 biological replicates.

Side comments

Because many figures and arguments in the paper discuss the percentage bound, the fraction stably bound and the stable binding time a more comprehensive description of how those measurements represent biologically and how they are extracted from the SPT data would be a nice addition.

In the revised manuscript we have we have now substantially expanded the initial description of these experiments and how the data presented were derived from them (lines 146-155 and 157-160). We have also attempted to highlight what they represent biologically where possible.

In Figure 8H, a legend should be added next to the graph

We thank the reviewer for pointing out this oversight and we have now added a legend to Figure 8H.

The legends of Figure S1 A and B are inverted.

We apologise for this oversight and have now corrected the legends for Figure S1A and B.

Serge Pelet

Thank you, Serge, for your helpful and constructive comments.

References

1. Marashi, F. *et al.* Histone proteins in HeLa S3 cells are synthesized in a cell cycle stage specific manner. *Science* **215**, 683–685 (1982).
2. Bravo, R. & Celis, J. E. A search for differential polypeptide synthesis throughout the cell cycle of HeLa cells. *J. Cell Biol.* **84**, 795–802 (1980).
3. Milcarek, C. & Zahn, K. The synthesis of ninety proteins including actin throughout the HeLa cell cycle. *J. Cell Biol.* **79**, 833–838 (1978).
4. Spalding, J., Kaijawa, K. & Mueller, G. An extracted basic protein isolated from HeLa nuclei and resolved by electrophoresis. *Proc Nat Acad Sci USA* **56**, 1535–1542 (1966).
5. Mazza, D., Abernathy, A., Golob, N., Morisaki, T. & McNally, J. G. A benchmark for chromatin binding measurements in live cells. *Nucleic Acids Res.* **40**, 1–13 (2012).
6. Isono, K. *et al.* SAM domain polymerization links subnuclear clustering of PRC1 to gene silencing. *Dev. Cell* **26**, 565–577 (2013).
7. Ren, X., Vincenz, C. & Kerppola, T. K. Changes in the distributions and dynamics of polycomb repressive complexes during embryonic stem cell differentiation. *Mol. Cell Biol.* **28**, 2884–95 (2008).
8. Saurin, A. J. *et al.* The human polycomb group complex associates with pericentromeric heterochromatin to form a novel nuclear domain. *J. Cell Biol.* **142**, 887–98 (1998).
9. Wani, A. H. *et al.* Chromatin topology is coupled to Polycomb group protein subnuclear organization. *Nat. Commun.* **7**, 10291–10291 (2016).
10. Fursova, N. A. *et al.* Synergy between Variant PRC1 Complexes Defines Polycomb-Mediated Gene Repression. *Mol. Cell* **74**, 1020-1036.e8 (2019).
11. Riising, E. M. *et al.* Gene Silencing Triggers Polycomb Repressive Complex 2 Recruitment to CpG Islands Genome Wide. *Mol. Cell* **55**, 347–360 (2014).
12. Tavares, L. *et al.* RYBP-PRC1 complexes mediate H2A ubiquitylation at polycomb target sites independently of PRC2 and H3K27me3. *Cell* **148**, 664–678 (2012).
13. Dobrinić, P., Szczurek, A. T. & Klose, R. J. PRC1 drives Polycomb-mediated gene repression by controlling transcription initiation and burst frequency. *bioRxiv* 2020.10.09.333294 (2020) doi:10.1101/2020.10.09.333294.
14. Blackledge, N. P. *et al.* PRC1 Catalytic Activity Is Central to Polycomb System Function. *Mol. Cell* **77**, 857-874.e9 (2020).
15. Tamburri, S. *et al.* Histone H2AK119 Mono-Ubiquitination Is Essential for Polycomb-Mediated Transcriptional Repression. *Mol. Cell* **77**, 840-856.e5 (2020).
16. Zhen, C. Y. *et al.* Live-cell single-molecule tracking reveals co-recognition of H3K27me3 and DNA targets polycomb Cbx7-PRC1 to chromatin. *eLife* **5**, 1–36 (2016).
17. Lee, H. G., Kahn, T. G., Simcox, A., Schwartz, Y. B. & Pirrotta, V. Genome-wide activities of Polycomb complexes control pervasive transcription. *Genome Res.* **25**, 1170–1181 (2015).
18. Kimura, H. & Cook, P. R. Kinetics of core histones in living human cells: little exchange of H3 and H4 and some rapid exchange of H2B. *J. Cell Biol.* **153**, 1341–53 (2001).
19. Hansen, A. S. *et al.* Robust model-based analysis of single-particle tracking experiments with Spot-On. *eLife* **7**, (2018).
20. Mangel, W. F., McGrath, W. J., Xiong, K., Graziano, V. & Blainey, P. C. Molecular sled is an eleven-amino acid vehicle facilitating biochemical interactions via sliding components along DNA. *Nat. Commun.* **7**, 10202 (2016).
21. Lavis, L. D. *et al.* Bright photoactivatable fluorophores for single-molecule imaging. *Nat. Methods* **13**, 985–985 (2016).
22. Persson, F., Lindén, M., Unoson, C. & Elf, J. Extracting intracellular diffusive states and transition rates from single-molecule tracking data. *Nat. Methods* **10**, 265–269 (2013).
23. Bentley, M. L. *et al.* Recognition of UbH5c and the nucleosome by the Bmi1/Ring1b ubiquitin ligase complex. *EMBO J.* **30**, 3285–3297 (2011).
24. McGinty, R. K., Henrici, R. C. & Tan, S. Crystal structure of the PRC1 ubiquitylation module bound to the nucleosome. *Nature* **514**, 591–6 (2014).

REVIEWERS' COMMENTS

Reviewer #1 (Remarks to the Author):

In the revised manuscript, the authors addressed most of my points raised. The manuscript is ready for publication.

Two minor issues:

Line 843. What is t_1 in the formula?

Line 854. It might be better to express the formula in a similar way to the one on Line 843.

Reviewer #2 (Remarks to the Author):

The authors have addressed all of my comments.

One minor text change - the legend of Figure 6C and 6D needs to be updated to now mention CBX7.

Reviewer #3 (Remarks to the Author):

The authors have addressed my concerns.

Reviewer #4 (Remarks to the Author):

The authors have made a substantial effort to address all the points raised by the reviewers and I fully support the publication of this manuscript.

However, I'm still bothered by one of the issues raised by reviewer 3 and myself regarding the possibility that long binding events could be due to multiple proteins successively binding to the same chromatin structure. While the explanation provided by the authors regarding the low level of photo-activation is a very convincing argument. The intensity traces provided in the Reviewer Figure 3 do fully exclude this possibility because they are relatively noisy (for good reasons as explained by the authors). In addition, when viewing Supplementary Video 1 ($T=19.77$) and 4 ($T=7$), I have clearly the impression of seeing spots merging or splitting. Can the authors explain these behaviors? Otherwise, I would suggest that the authors mention the possibility of multiple successive binding events in the main text.

Minor comments

Some numbers are written with or without thousand separators.

I would recommend using the same scaling for plots with the same axis on the same figure for better comparison. For instance: Figure S5A ,and Figure S7A and B.

We would again like to thank the reviewers for their time and helpful input in revising our manuscript and their recommendation for publication. Below we have provided a point-by-point response (black text) to the reviewers' additional comments following revision (blue text).

Reviewer #1 (Remarks to the Author):

In the revised manuscript, the authors addressed most of my points raised. The manuscript is ready for publication.

Two minor issues:

Line 843. What is t_1 in the formula?

In the double exponential function used to fit the apparent dwell times, t_1 represents the first time point (i.e. 0 s). This has now been added to the methods text to further clarify this equation.

Line 854. It might be better to express the formula in a similar way to the one on Line 843.

The equations in question have been reformatted so that they are expressed in a consistent manner.

Reviewer #2 (Remarks to the Author):

The authors have addressed all of my comments.

One minor text change - the legend of Figure 6C and 6D needs to be updated to now mention CBX7.

We thank the reviewer for noticing this oversight. The legends in question have now been updated.

Reviewer #3 (Remarks to the Author):

The authors have addressed my concerns.

Reviewer #4 (Remarks to the Author):

The authors have made a substantial effort to address all the points raised by the reviewers and I fully support the publication of this manuscript.

However, I'm still bothered by one of the issues raised by reviewer 3 and myself regarding the possibility that long binding events could be due to multiple proteins successively binding to the same chromatin structure. While the explanation provided by the authors regarding the low level of photo-activation is a very convincing argument. The intensity traces provided in the Reviewer Figure 3 do fully exclude this possibility because they are relatively noisy (for good reasons as explained by the authors). In addition, when viewing Supplementary Video 1 ($T=19.77$) and 4 ($T=7$), I have clearly the impression of seeing spots merging or splitting. Can the authors explain these behaviors? Otherwise, I would suggest that the authors mention the possibility of multiple successive binding events in the main text.

We appreciate the reviewer's concern on this topic. It is worth noting that in 67 Hz experiments, which are more prone to tracking errors due to motion blur, individual missing frames are tolerated (which could occur if two molecules overlap such that they are localised as a single molecule on one frame). Additionally, since only part of each track is used in analysis, and the individual jumps are analysed as an overall population rather than as part of a track, single frame disruptions to tracks will have an overall very minor effect as long as they are infrequent (for example, Supplementary Movie 1 is half of a 60 s acquisition which contains a total of 4289 jumps).

For 2 Hz and 0.033 Hz experiments, overlapping molecules may pose a larger issue for determining dwell times of individual molecules. However, as the reviewer notes, analysis of intensity over time along tracks did not show any evidence of step-wise photobleaching, which would be expected if tracks consisted of >1 molecule. Furthermore, while the molecules in Supplementary Movie 4 appear close, they are still sufficiently distinct to be tracked separately. As such, while overlapping can occur, the probability of this occurring frequently enough in the acquired data to significantly influence the analysis is low.

To recognise that this possibility exists, we have now included a sentence in the manuscript (lines 150 – 152) acknowledging that we have tried to minimise this effect: "The acquisition of these movies was performed with limited photoactivation of molecules such that only a small number of molecules were visible at once. This helped to reduce the probability of tracks of multiple molecules overlapping and being conflated in our analysis."

Minor comments

Some numbers are written with or without thousand separators.

The use of thousand separators has now been made consistent throughout the manuscript.

I would recommend using the same scaling for plots with the same axis on the same figure for better comparison. For instance: Figure S5A ,and Figure S7A and B.

We agree with the reviewer's suggestion. The y-axes for box plots in Figure 8g and Supplementary Fig. 5a and 7a and b are now scaled identically to facilitate comparison.